# Cocktail🍸: Mixing Multi-Modality Controls for Text-Conditional Image Generation

Minghui Hu[†], Jianbin Zheng[⋆], Daqing Liu[‡], Chuanxia Zheng[§], Chaoyue Wang[¶], Dacheng Tao[¶], and Tat-Jen Cham[†]

[†]*Nanyang Technological University,* [⋆]*South China University of Technology,* [§]*University of Oxford,* [¶]*The University of Sydney,* [‡]*JD Explore Academy*

## Abstract

Text-conditional diffusion models are able to generate high-fidelity images with diverse contents. However, linguistic representations frequently exhibit ambiguous descriptions of the envisioned objective imagery, requiring the incorporation of additional control signals to bolster the efficacy of text-guided diffusion models. In this work, we propose Cocktail, a pipeline to mix various modalities into one embedding, amalgamated with a generalized ControlNet (gControlNet), a controllable normalisation (ControlNorm), and a spatial guidance sampling method, to actualize multi-modal and spatially-refined control for text-conditional diffusion models. Specifically, we introduce a hyper-network gControlNet, dedicated to the alignment and infusion of the control signals from disparate modalities into the pre-trained diffusion model. gControlNet is capable of accepting flexible modality signals, encompassing the simultaneous reception of any combination of modality signals, or the supplementary fusion of multiple modality signals. The control signals are then fused and injected into the backbone model according to our proposed ControlNorm. Furthermore, our advanced spatial guidance sampling methodology proficiently incorporates the control signal into the designated region, thereby circumventing the manifestation of undesired objects within the generated image. We demonstrate the results of our method in controlling various modalities, proving high-quality synthesis and fidelity to multiple external signals. The codes are released at https://mhh0318.github.io/cocktail/.

## 1 Introduction

Text-conditional diffusion models [41, 39, 42, 34] have actualized the capacity for high-quality generative capabilities. These models facilitate the generation of an array of high-calibre images through the utilization of concise textual prompts. However, linguistic representations pose inherent challenges in accurately encapsulating the precise imagery anticipated by the user, owing to the potential ambiguities and subjective interpretations of verbal descriptions within the context of visual synthesis. Moreover, minor alterations to the textual prompts also yield distinct visual outputs, underscoring the absence of refined control over the generative process.

Modifying the prior is a series of existing solutions for multi-modal control, *e.g.*, the control of the entire prior space [39, 48, 23, 22, 57] as demonstrated in Fig. 1(a). These approaches centered on the whole prior lack the capacity for localised image modifications and the preservation of background elements. Moreover, these models typically require training from scratch, which demands a substantial amount of resources.

37th Conference on Neural Information Processing Systems (NeurIPS 2023).

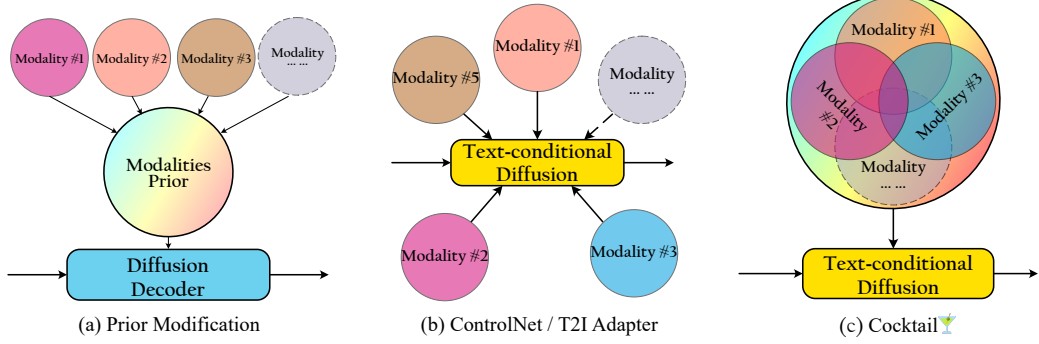

(a) Prior Modification     (b) ControlNet / T2I Adapter     (c) Cocktail🍸

Figure 1: **Comparison of various control methods.** Our approach requires only *one generalized model*, unlike previous that needed multiple models for multiple modalities.

In response to the methods dealing with latent representation [29, 4, 27], an additional lightweight hyper-network is introduced in [53, 33], which is designed to encode external control signals into latent vectors and subsequently inject them directly into the backbone network. Such methods effectively handle control over the single additional modality; however, they exhibit limitations when confronted with multiple modalities, as shown in Fig 1(b). One issue is that each modality requires a unique network, leading to a computational overhead that escalates proportionally with the increase in the number of modalities. Furthermore, the impact of additive coefficients between different modes on the final imaging outcomes warrants consideration. The inherent imbalance among superimposed modes makes these additive coefficients a pivotal factor in determining the ultimate synthesized output. An additional challenge emerges during the sampling process when the model conducts an initial inference devoid of control signal injection. This preliminary inference step could result in object placement that potentially contradicts the control signals.

In this paper, we propose a novel pipeline, as shown in Fig. 1(c), termed Cocktail, which accomplishes multi-modality control through a text-conditional diffusion model. It encompasses three main components: **1)** a hyper-network capable of accommodating multi-modal input, **2)** a conditional normalisation method to mix the control features, and **3)** a sampling strategy designed to facilitate precise control over the generative process. As shown in Fig. 2, our method is proficient in generating images that meet all input conditions or any arbitrary subset thereof, *utilizing only one single model*.

To achieve this goal, we initially trained a branched network named gControlNet, which accepts multiple modalities. Upon training, the branched network can simultaneously accept arbitrary combinations of existing modalities. When multiple modalities coexist within the same region, the model is capable of automatically fusing input from different modalities, balancing the disparities between them. To leverage the features of the gControlNet, we further proposed controllable normalisation (ControlNorm). We can achieve better representation of control signals in terms of semantic and spatial aspects according to the decoupling provided by ControlNorm.

Moreover, we introduce a spatial guidance sampling method in order to facilitate spatial generation under the purview of multi-modal control signals. Specifically, we employ distinct textual prompts to differentiate entities from the background, incorporating entities into the background via a prompt editing approach. Our devised sampling approach demonstrates efficacy in circumventing the generation of undesired entities.

In summary, our main contributions are as follows:

- We introduced the Generalized ControlNet (gControlNet), a branched network capable of adaptively integrating multi-modal information, effectively addressing issues stemming from imbalances between modalities;

- We proposed the Controllable Normalistion (ControlNorm) to optimize the utilization of information within branched networks to yield more effective outcomes;

- We introduced a spatial guidance sampling method based on the operation within the attention map to generate relevant information tailored to regional contexts, preventing the inclusion of undesired objects outside the specified regions.

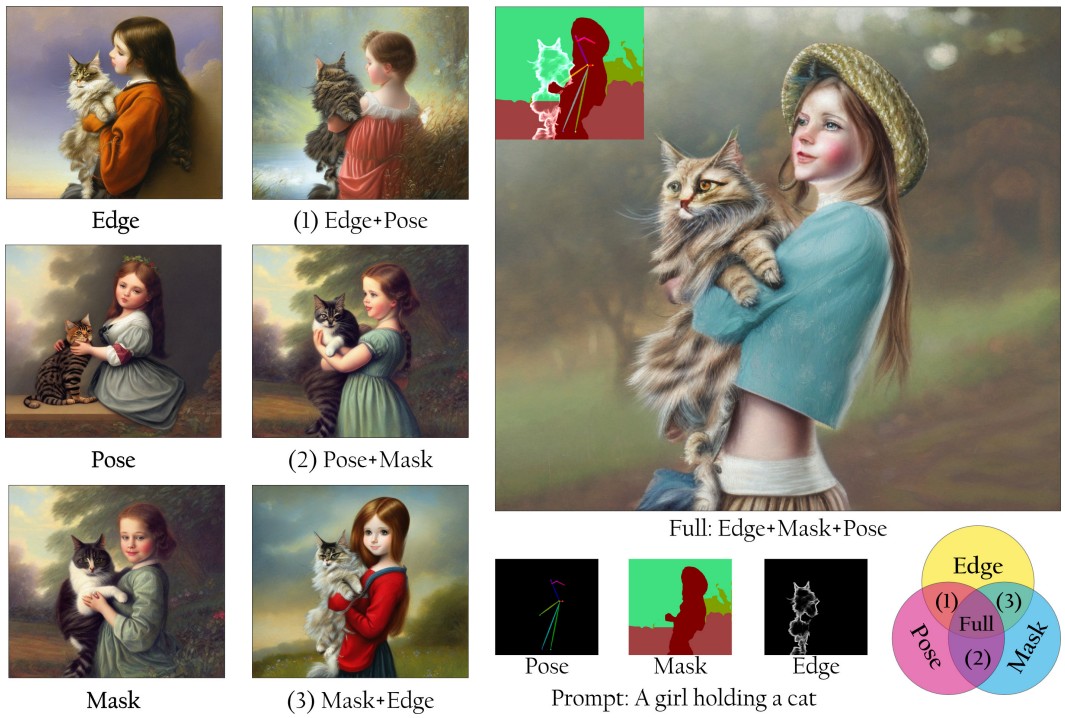

Figure 2: **Examples of our model with the same prompt.** Given a text prompt along with various modality signals, our approach is able to synthesize images that satisfy *all input conditions* or *any arbitrary subset* of these conditions using *a single model*. The prompt is: *A girl holding a cat.*

## 2 Related Work

**Text-conditional Diffusion Models.** Diffusion models [19, 46] has achieved great success in the area of text-to-image synthesis [34, 39, 41, 3, 14]. To reduce the computational cost, diffusion models typically function within the latent space [41] or produce low-resolution images that are later improved through super-resolution models [39, 3]. Fast sampling methods have also successfully reduced the number of generation steps required by diffusion models from hundreds down to merely a few [45, 32, 25, 30, 11]. The provision of classifier guidance during the sampling process can also significantly impact the outcomes, leading to substantial improvements in the results [9]. In addition to the widely used classifier-free guidance [18], other types of guidance are also worth exploring [55, 13].

**HyperNetworks for Pre-trained Models.** Training a diffusion model is highly resource-intensive and environmentally unfriendly [41]. Fine-tuning such a model can also be challenging due to the vast number of parameters involved [1]. Therefore, introducing an additional tiny branched network to bias the output of the original network is a more reasonable choice [10, 2, 8]. Similar ideas have also proven to be effective in diffusion models: Hypernet [15] and LoRA [20] models are capable of altering the original diffusion model's sampling distribution by training a small branched network, which has become the most popular branched network in the current research community. More similar to our work is ControlNet [53], which is subsequently combined with different layers in the denoising U-Net to provide support for various task-specific guidance. It presents impressive results with various conditional inputs, *yet with only one modality for each model.* In contrast, our cocktail endows the ControlNet with multitasking capabilities using only *one single model.*

**Conditional Normalisation** approaches have been employed across a range of vision tasks, including style transfer [24, 12], conditional generation [36, 7, 14, 21] and image-to-image translation [40]. These techniques involve normalizing layer activations to zero mean and unit deviation, followed by denormalisation through an affine transformation derived from external data. External data can be in multiple formats, such as style images, semantic masks, or category labels.

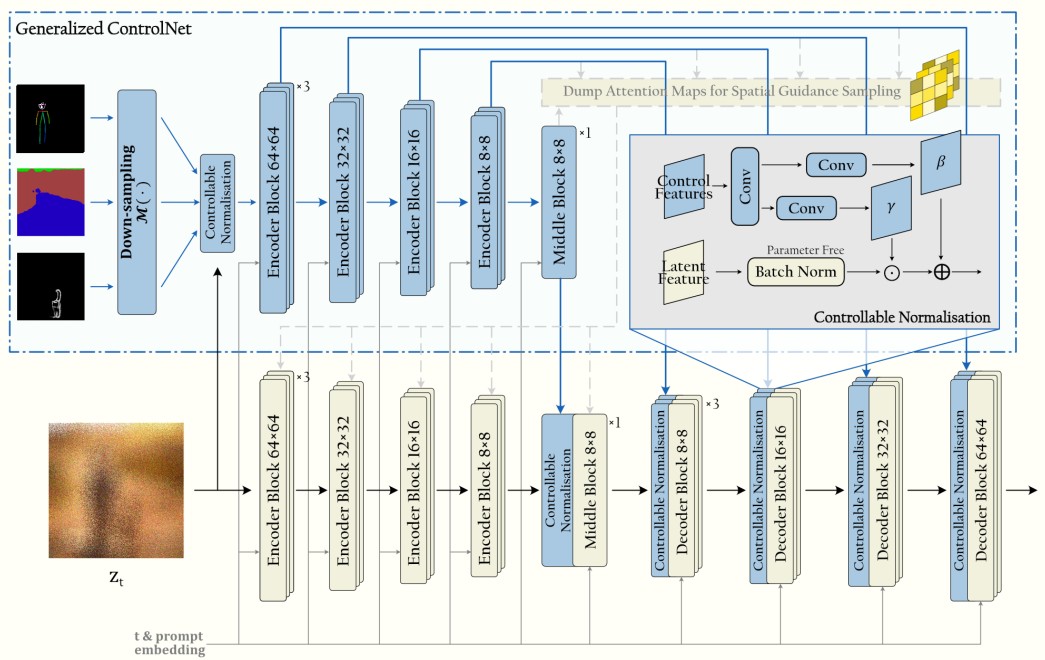

**Figure 3: The network architecture** of Generalized ControlNet (gControlNet) with Controllable Normalisation (ControlNorm). The parameters indicated by the yellow sections are sourced from the pre-trained model and stay constant, while only those in the blue sections are updated during training, with the gradient back-propagated along the blue arrows.

**Attention Map-based Prompt Tuning.** Prompt-to-Prompt [16] is a method that adjusts local or global specifics in text-guided diffusion models by altering the cross-attention maps from source to target image, thereby maintaining spatial layout and geometry. Recently, numerous efforts have been made to improve the outcomes [35, 37]. However, such approaches are limited to synthesized images without an inversion technique. Unlike cross-attention, self-attention focuses on inter-pixel relationships within the same domain [47]. Paint-with-words [3] is another method that allows users to specify the spatial locations of objects by selecting phrases from the text prompt.

## 3 Methods

In this work, our *main goal* is to design a controllable generator that utilizes *various modalities* of input, within a *single model*. To achieve this, we first propose a more general branched network, gControlNet, which can generate control signals from different modalities using a single network and adaptively weighted fuse them together. Furthermore, we propose ControlNorm to inject the signals from the gControlNet into the diffusion backbone, which solves the imbalanced problem within various modalities. Finally, we propose a spatial guidance sampling method to avoid the presence of extraneous objects in the generation process. By modifying the attention map, this approach effectively incorporates control signals into the backbone network. The whole pipeline is demonstrated in Fig. 3.

### 3.1 Generalized ControlNet with Controllable Normalisation

**Generalized ControlNet.** ControlNet [53] is a method designed to influence and control the behavior of neural networks by adjusting the input conditions of specific network blocks. Instead of directly modifying the parameters of the primary network, ControlNet employs an auxiliary network to generate feature offsets. These offsets are then combined with different layers in the main network, such as a denoising U-Net, to support various task-specific guidance.

Given a trained backbone network block $\mathcal{F}(\cdot; \boldsymbol{\theta})$ with parameter $\boldsymbol{\theta}$, the input feature $\boldsymbol{x}$ can be mapped to $\boldsymbol{y}$. For the branched part, we duplicate the parameter $\boldsymbol{\theta}$ to create a trainable copy $\boldsymbol{\theta}_t$, which is

then trained using the supplementary modality. Preserving the original weights helps retain the information stored in the initial model after training on large-scale datasets, which ensures that the quality and diversity of the generated images do not degrade. Mathematically, the output from the trained network block can be expressed as:

$$\boldsymbol{y} = \mathcal{F}(\boldsymbol{x}; \boldsymbol{\theta}) + \mathcal{Z}\left(\mathcal{F}(\boldsymbol{x} + \mathcal{Z}(\boldsymbol{c}_m); \boldsymbol{\theta}_t)\right) \leftarrow \mathcal{F}(\boldsymbol{x}; \boldsymbol{\theta}), \tag{1}$$

where the control signal of a single modality $\boldsymbol{c}_m$ is typically processed to obtain a format identity for $\boldsymbol{x}$, for example, through a zero-initialized convolutional layer, and then added to $\boldsymbol{x}$. $\mathcal{Z}(\cdot)$ represents the zero-initialized layer. It is not only used in the process of handling $\boldsymbol{c}_m$, but also serves to adjust the output of the branched network $\mathcal{F}(\cdot, \boldsymbol{\theta}_t)$.

To accomplish the goals of accepting multiple external modalities as input and balancing signals from different modalities, we have devised a modified framework that adeptly merges these varied sources of information. At the top of our network, we adopt a simple downsampling network $\mathcal{M}(\cdot)$ to convert external conditional signals to the latent space, allowing the conditional signals to be directly injected into the latent space. It is worth noting that $\mathcal{M}(\cdot)$ is versatile and can adapt to different types of external signals. Given $k$ different modalities, the converted conditional features are $\boldsymbol{c}_m^k = \mathcal{M}(C^k)$.

**Controllable Normalisation.** Instead of directly passing the sum of conditional features via a zero-initialized layer to the network block $\mathcal{F}(\cdot; \boldsymbol{\theta}_t)$, i.e., $\hat{\boldsymbol{c}}_m = \mathcal{Z}(\sum_i \boldsymbol{c}^i)$, we introduce a *controllable normalisation (ControlNorm)* method, which has an additional layer to generate two sets of learnable parameters, $\boldsymbol{\gamma}(\hat{\boldsymbol{c}}_m)$ and $\boldsymbol{\beta}(\hat{\boldsymbol{c}}_m)$, conditioned on all $k$ modalities. These two sets of parameters are used in the conditional normalisation layer to fuse the external conditional signals and the original signals. Specifically, the input to the trainable block $\mathcal{F}(\cdot; \boldsymbol{\theta}_t)$ becomes:

$$(\boldsymbol{I} + \mathcal{Z}(\boldsymbol{\gamma}\left(\hat{\boldsymbol{c}}_m\right))) \odot \frac{\boldsymbol{x} - \mu_c(\boldsymbol{x})}{\sigma_c(\boldsymbol{x})} \oplus \mathcal{Z}(\boldsymbol{\beta}(\hat{\boldsymbol{c}}_m)) \leftarrow \boldsymbol{x} + \mathcal{Z}(\boldsymbol{c}_m), \tag{2}$$

where $\mu_c(\boldsymbol{x})$ and $\sigma_c(\boldsymbol{x})$ are the mean and standard deviation of the feature $\boldsymbol{x}$ along the channel $c$, $\odot$ is Hadamard product, and $\boldsymbol{\gamma}(\hat{\boldsymbol{c}})$ and $\boldsymbol{\beta}(\hat{\boldsymbol{c}})$ are vectors that have the same dimension as $\boldsymbol{x}$. With the help of zero-convolution, we can preserve the identity of $x$ just before the start of the fine-tuning. It is worth noting that in the following layers, the internal feature in the branched network $\boldsymbol{h}$ can be integrated with the latent feature $\boldsymbol{z}$ from the original network in the same way:

$$(\boldsymbol{I} + \mathcal{Z}(\boldsymbol{\gamma}\left(\boldsymbol{h}\right))) \odot \frac{\boldsymbol{z} - \mu_c(\boldsymbol{z})}{\sigma_c(\boldsymbol{z})} \oplus \mathcal{Z}(\boldsymbol{\beta}(\boldsymbol{h})) \leftarrow \boldsymbol{x} + \mathcal{Z}(\boldsymbol{c}_m), \tag{3}$$

where $\boldsymbol{h}$ is the intermediate features from the original network and $\boldsymbol{z}$ is the intermediate features from the branched network. Specifically, we will have five sets of intermediate features from the Stable Diffusion [41] U-Net backbone and our generalized ControlNet, including four sets of features from encoder blocks and one from the middle block.

In fact, our controllable normalisation is a generalized version of conditional normalisation [36, 24]. After changing the mean and variance calculation dimension and replacing the external signal $\hat{\boldsymbol{c}}$ by a mask image, real image, or class labels, we can derive the various forms of SPADE [36], AdaIN [24], CIN [12] and MoVQ [56]. More interestingly, our controllable normalisation method not only enables the use of external signals as conditions, but also allows intermediate-layer signals to act as constraints.

As shown in Fig. 3, only the parameters of the gControlNet require updating. Given an initial latent $\boldsymbol{z}_0$ and a time-step $t$, the diffusion process progressively introduces noise $\epsilon$ to this latent, transforming the original latent into a noisy state $\boldsymbol{z}_t$. The objective of the diffusion model is to estimate the noise $\epsilon$ adding to the $\boldsymbol{x}$. Our proposed gControlNet shares the same objective function as the diffusion model, aiming to predict the noise added at time $t$. The only distinction lies in the incorporation of multimodal information as conditional signals $\hat{\boldsymbol{c}}_m = \left[\boldsymbol{c}_m^0, \boldsymbol{c}_m^1, \ldots, \boldsymbol{c}_m^k\right]$:

$$\mathcal{L} = \mathbb{E}_{\boldsymbol{z}_0, t, \boldsymbol{c}_p, \hat{\boldsymbol{c}}_m, \epsilon \sim \mathcal{N}(0,1)}\left[\|\epsilon - \epsilon_\theta\left(\boldsymbol{z}_t, t, \boldsymbol{c}_p, \hat{\boldsymbol{c}}_m\right)\|_2^2\right] \tag{4}$$

## 3.2 Spatial Guidance Sampling

In order to leverage the control signals from generalized ControlNet and ensure that the generated objects appear within the areas of interest, we proposed a *spatial guidance* sampling method. Here, we mainly focus on editing the cross-attention layers of the U-Net in Stable Diffusion.

Given a noisy latent $\boldsymbol{z}$ at timestep $t$ as the input to the pretrained U-Net with $n$ blocks $\mathcal{F}^{(n)}(\cdot; \boldsymbol{\theta}^{(n)})$ at different resolution, the latent vector $\boldsymbol{z}$ will be down-sampled to various dimensional $\boldsymbol{z}^{(n)}$ that map to the corresponding block $\mathcal{F}^{(n)}$. The cross-attention maps $\boldsymbol{A}^{(n)} \in \mathbb{R}^{(N_i, N_t)}$ in each block, parameterized by $\boldsymbol{\theta}^{(n)}$, are associated with the linear projection of latent feature $Q^{(n)} = f_Q(\phi(\boldsymbol{z}^{(n)}))$ and the prompt $K = f_K(\boldsymbol{c}^{\text{text}})$:

$$A_{ij|\boldsymbol{\theta}^{(n)}}^{(n)} = \frac{\exp\langle Q_i^{(n)}, K_j\rangle}{\sum_{k=1}\exp\langle Q_i^{(n)}, K_k\rangle}, \tag{5}$$

where $A_{ij}^{(n)}$ represents the attentional strength between the $j$-th prompt token (among $N_t$ tokens) and the $i$-th latent feature (among $N_i$ features). Intuitively, for each prompt token $K_j$, there exists a corresponding latent feature map $A_j^{(n)}$ with spatial information. Moreover, the corresponding feature map will contain spatial and shape information for the associated object. However, it is not only the object-describing tokens that contain information about the object's location and shape, some connecting words and padding tokens also convey spatial information for the overall scene [6, 43].

We apply a masking strategy to the corresponding attention maps. In detail, we construct two sets of attention masks $M^{\text{pos}(n)}$ and $M^{\text{neg}(n)} \in \mathbb{R}^{(N_i, N_t)}$. Each column $M_j^{\text{pos}(n)}$ and $M_j^{\text{neg}(n)}$ is a flattened alpha mask, which is determined by the visibility of the corresponding text token $K_j$. The values of $M_{ij}^{\text{pos}(n)}$ and $M_{ij}^{\text{neg}(n)}$ are determined based on the relationship between image token $Q_i$ and text token $K_j$. On the one hand, if image token $Q_i$ corresponds to a region of the image that should be influenced by text token $K_j$, $M_{ij}^{\text{pos}(n)}$ is assigned the value of 1. On the other hand, if image token $Q_i$ corresponds to a region of the image that should not be influenced by text token $K_j$, $M_{ij}^{\text{neg}(n)}$ is set to 1. It is worth noting that we consider the feature maps corresponding to most words as negative in order to avoid generating undesired objects. The mask components $M^{\text{pos}(n)}$ and $M^{\text{neg}(n)}$ are incorporated into the cross-attention computation process:

$$\tilde{A}_{ij|\boldsymbol{\theta}^{(n)}}^{(n)} = \frac{\exp\langle Q_i^{(n)}, K_j\rangle + \omega^{\text{pos}}M^{\text{pos}(n)} - \omega^{\text{neg}}M^{\text{neg}(n)}}{\sum_{k=1}\exp\langle Q_i^{(n)}, K_k\rangle}. \tag{6}$$

It is found that larger weights at higher noise levels [3] can lead to better results, thus $\omega^{\text{pos}}$ and $\omega^{\text{neg}}$ are noise-level sensitive parameters, defined by:

$$\omega^{(\cdot)} = \omega' \cdot \log(1 + \sigma) \cdot \max(\boldsymbol{A}^{(n)}), \tag{7}$$

where $\omega'$ is a user-provided hyper-parameter.

We then substitute the feature map corresponding to the object description. Contrary to the image editing motivation behind conventional prompt tuning methods, it is not necessary for our method to provide a reference image and an amended prompt. Recalling the framework of our generalized ControlNet, the branch architecture is identical to the encoder portion of the backbone network. We found that the attention maps $A_{j|\boldsymbol{\theta}_t^{(n)}}^{(n)}$ within the branch network also encompass object locations and shape information. Consequently, we opt for the attention map $A_{j|\boldsymbol{\theta}_t^{(n)}}^{(n)}$ associated with the description $K_j$ from the respective layer in ControlNet as the source for the original attention map $A_{j|\boldsymbol{\theta}^{(n)}}^{(n)}$ substitution, ensuring that the information in the substituted attention map aligns more closely with the external input signal rather than the textual information derived from the original backbone network. Specifically, we replace the attention map generated by the original backbone network (based on the corresponding object description) with the attention map from the corresponding module in the branch network:

$$\hat{\boldsymbol{A}}^{(n)} = [\tilde{A}_{0|\boldsymbol{\theta}^{(n)}}^{(n)}; \ldots; A_{j|\boldsymbol{\theta}_t^{(n)}}^{(n)}; \ldots; \tilde{A}_{N_t|\boldsymbol{\theta}^{(n)}}^{(n)}] \leftarrow [\tilde{A}_{0|\boldsymbol{\theta}^{(n)}}^{(n)}; \ldots; \tilde{A}_{j|\boldsymbol{\theta}^{(n)}}^{(n)}; \ldots; \tilde{A}_{N_t|\boldsymbol{\theta}^{(n)}}^{(n)}]. \tag{8}$$

Subsequently, we can produce the spatially guided output from cross-attention layer by taking the product of $\hat{\boldsymbol{A}}^{(n)}$ with $V$.

# 4 Experiments

In this section, we delve into a comprehensive experimental analysis to validate the efficacy and superiority of the proposed method through ablation studies and application demonstrations. Subse-

Table 1: Quantitative comparison on the COCO5k validation set. The best result is highlighted.

| Method | Similarity (LPIPS ↓) | Sketch Map (L2 Distance ↓) | Segmentation Map (mPA ↑) | Segmentation Map (mIoU ↑) | Pose Map (mAP ↑) |
|---|---|---|---|---|---|
| Multi-Adapter | 0.7273 $\pm 0.00120$ | 7.93310 $\pm 0.01392$ | 26.30 $\pm 0.242$ | 13.98 $\pm 0.177$ | 40.02 $\pm 0.761$ |
| Multi-ControlNet | 0.6653 $\pm 0.00145$ | 7.59721 $\pm 0.01516$ | 36.59 $\pm 0.273$ | 22.70 $\pm 0.229$ | 38.19 $\pm 0.761$ |
| *Ours* w/o ControlNorm | 0.4900 $\pm 0.00141$ | **7.18413** $\pm 0.01453$ | 48.26 $\pm 0.287$ | 32.66 $\pm 0.272$ | 61.93 $\pm 0.775$ |
| *Ours* 🍸 | **0.4836** $\pm 0.00133$ | 7.28929 $\pm 0.01385$ | **49.20** $\pm 0.289$ | **33.27** $\pm 0.271$ | **61.99** $\pm 0.778$ |

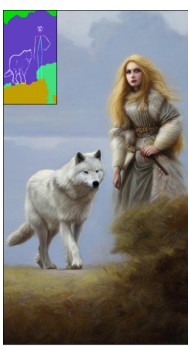 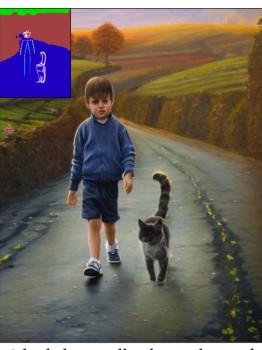 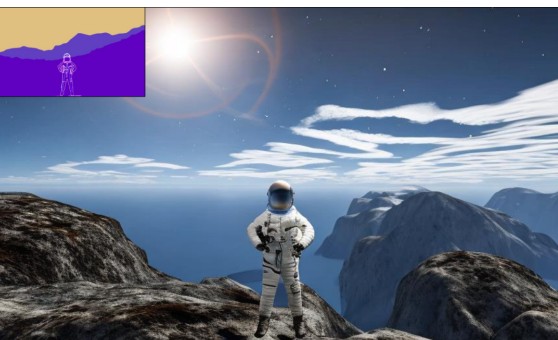

A huntress and a big white wolf.     A little boy walks down the road, followed by a cat.     An astronaut standing on the mountain

Figure 4: Our model can generate images with the provided prompts and multi-modality information (e.g., edge, pose, and segmentation map) across various scales.

quently, in Sec. 4.1, we put forth both quantitative and qualitative results, elucidating the comparative advantages of our approach. We also present an array of intriguing applications made possible by our gControlNet, showcasing its practical utility. Finally, Sec. 4.2 is dedicated to the discussion of ablation studies, scrutinising the impacts of varying injection and sampling methods. The experimental configurations, including the dataset specifications, implementation details, and evaluation metrics, can be found in the Appendix.

## 4.1 Applications and Comparisions

Cocktail is proficient in seamlessly supporting multiple control inputs and autonomously fusing them, thereby eliminating the necessity for manual intervention to equilibrate diverse modalities. This unique property empowers users to easily incorporate a variety of modalities, resulting in more flexible multi-modal control. In Figure 4, we demonstrate how users can provide spatial information about multiple objects in different modalities to generate a complex scene. Notably, this entire process is accomplished by a single model without the need for additional branch networks.

We then compare Cocktail with two state-of-the-art methods: ControlNet [53] and T2I-Adapter [33], in the context of text-guided image-to-image translation within multiple modalities. We employ several evaluation methods, including LPIPS, mPA, mIoU, mAP and L2 distance for various modalities in this section. As depicted in Table 1, our method outperformed both ControlNet and T2I-Adapter across all evaluation metrics. This remarkable achievement signifies that our proposed cocktail can generate a structural image that closely resembles the ground truth image and aligns better with the input conditions, establishing its superiority. The visualization in Fig. 6 also illustrates the superior ability of our model to harmonize with the control signals.

We further present some samples from our method to examine its effectiveness on uni-modality translation. The visual comparison on the COCO validation set is showcased in Figure 5, highlighting the compelling performance. Benefiting from mixed training of multiple modalities, our method achieves remarkable generation quality, surpassing even models exclusively trained for a single control signal.

Our experiments show that Cocktail effectively leverages information from different modalities and exhibits outstanding multi-object generation abilities with consistent composition across various control signals.

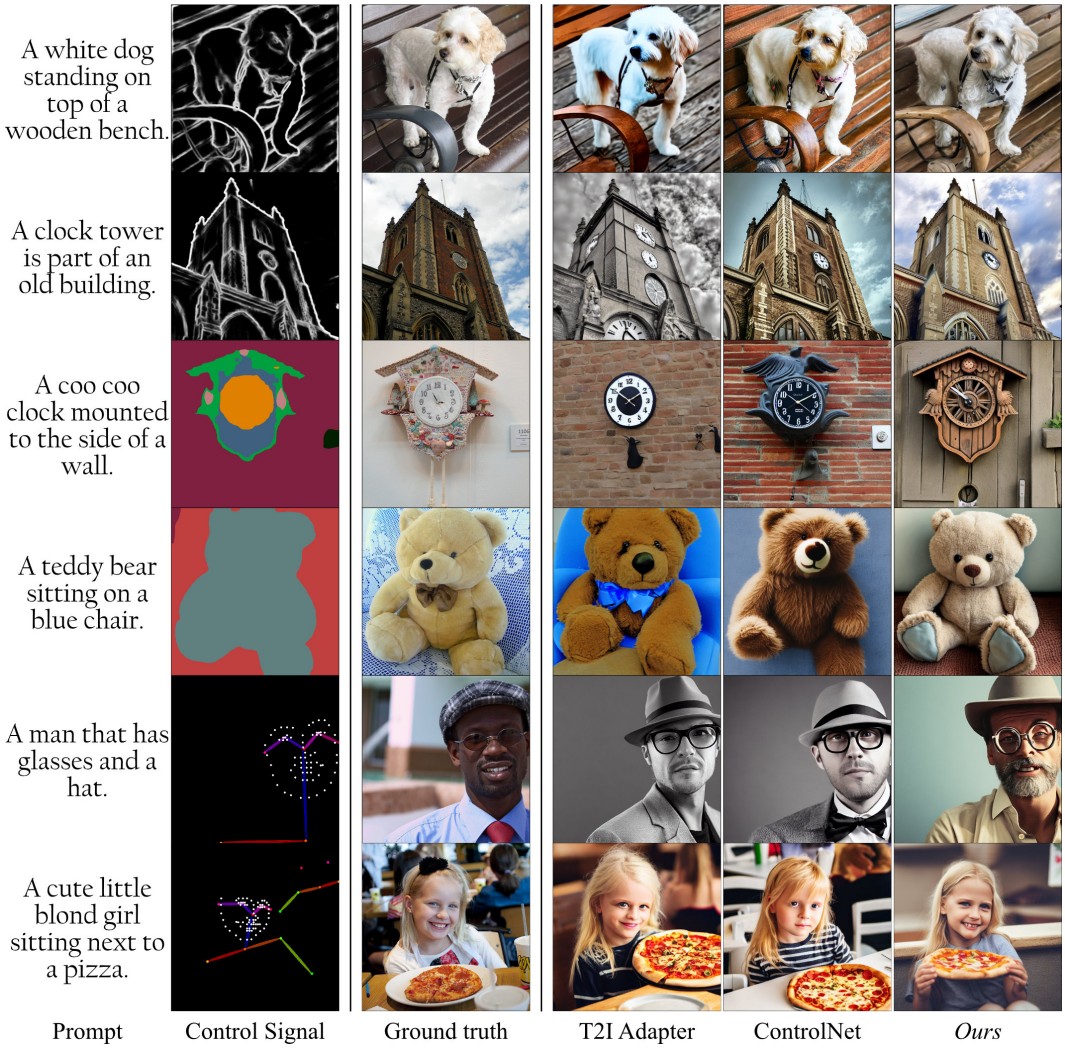

| Prompt | Control Signal | Ground truth | T2I Adapter | ControlNet | *Ours* |

Figure 5: Qualitative comparison of Uni-Modality on the COCO validation set.

## 4.2 Ablations

**gControlNet and ControlNorm**   Previous methods [53] utilized a direct sum approach to fuse control signals from hyper-networks with latent variables from the original network. While this method conveys spatial information, it fails to consider semantic information, such as text or other modalities. Control signals decoupled through ControlNorm allow the preservation of semantic information while conveying spatial information. Another point is the need for normalization to address the imbalance of different modality signals. However, typical normalization methods lead to the loss of semantic information [36]. Therefore, the control signals introduced through ControlNorm can better interpret conditional information. We present some generated images in Fig. 10 to substantiate the interpretative capability of ControlNorm.

**Spatial Guidance Sampling**   The spatial guidance sampling method not only ensures that objects are generated within controllable areas, but also minimizes the impact on other areas. An intuitive application is that when we modify certain objects or modalities, other parts of the generated image can remain unchanged. Figure 7 illustrates the contrast between employing spatial guidance sampling and its absence. A more significant variation in the overall tone of the resultant image is observed when spatial guidance is not utilized, leading to inconsistencies in the details of the attire. Conversely, the incorporation of new objects and modalities with the use of spatial guidance minimally impacts the original image.

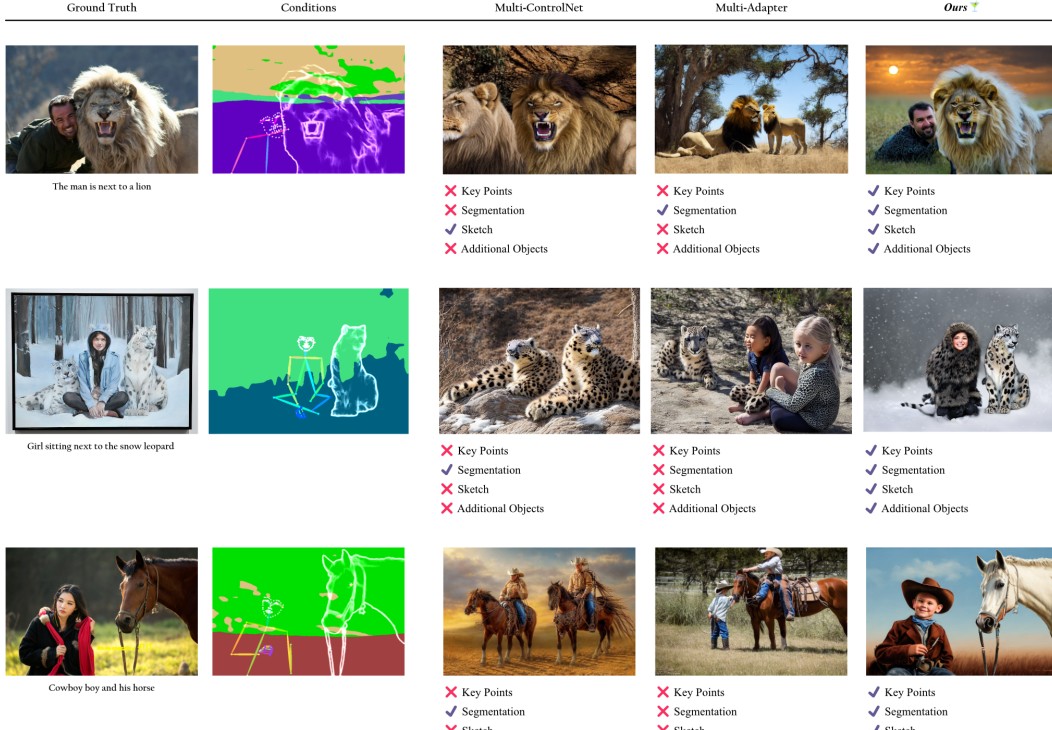

Figure 6: Cocktail can address the imbalance among various modalities. Here, the "cross" symbol and the checkmark symbol denote the unmatched and matched modalities, respectively. It is important to note that our model accurately captures all modalities.

## 5 Conclusion

In this work, we proposed Cocktail, a pipeline to achieve fine control by fusing multi-modal signals. Firstly, we introduced a generalized ControlNet capable of handling signals of different modalities using a single model. We also revealed that the semantic information of signals from the hyper-network may be lost through direct addition. However, a simple decoupling allows for a better interpretation of control signals. We introduced a controllable normalisation for decoupling and integration. Finally, we proposed a sampling scheme that can prevent the generation of unnecessary objects outside the focus area while also achieving a certain degree of editing capability and background protection.

**Limitations.** While we have achieved multi-modal fusion and control, there are still certain limitations in the handling of control signals from Cocktail that need to be addressed in future work. Firstly, although the current spatial guidance method can effectively prevent objects from being generated outside the focus area, it requires users to individually specify the area and corresponding object description during implementation. Secondly, spatial guidance can cause instability in the latent space in certain situations, leading to the generated images degrading and deviating from the existing control signals. Therefore, finding a stable anchor point as a reference during the generation process is also worth exploring.

**Broader Impacts.** The integration of multimodal control signals as inputs for synthesis greatly enhances user interaction flexibility and streamlines the utilization of text-conditional diffusion models. However, the process of fine-tuning large-scale generative models to accommodate diverse modalities requires a significant amount of energy, and we anticipate that the development of a universal model capable of accepting multiple modalities will help mitigate this impact. However, the growing capabilities in image generation also facilitate the production of manipulated images with malicious intent, such as the creation of counterfeit or deceitful information.

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

# A  Experimental settings

## A.1  Datasets and Annotations.

All of our experiments are performed on LAION-AESTHETICS-6.5 dataset, which contains about 600K image-text pairs with predicted aesthetics scores of higher than 6.5.

Our experiment includes three types of commonly used additional control signals, and we use different methods to obtain pseudo-labeling annotations:

- *Sketch map*: The delineation of texture details within the structured control signal is accomplished using the HED boundary detector [50]
- *Semantic segmentation map*: To derive segmentation maps from images, we leverage SAN [52], a high-performing, open-vocabulary segmentator.
- *Human pose map*: The generation of a comprehensive human pose map, including body, face, and hand positions, is facilitated by OpenPose [5].

It is crucial to note that our proposed Generalized ControlNet exhibits broad applicability and can be effortlessly adapted to accommodate a multitude of other modality inputs.

## A.2  Implementation Details.

gControlNet is adapted from the pretrained Stable Diffusion v2.1 in this paper and trained for 20 epochs with a batch size of 64 on 4 NVIDIA 80G-A100 GPUs within 4 days. We use the AdamW optimizer with a learning rate of 3.0 e-05. All the training images in the LAION-AESTHETICS-6.5 are first resized to 512 by the short side and then randomly cropped to $512 \times 512$. During inference, the sampler is DDIM, the sampling steps are 50, and the classifier-free guidance scale is 9.0 by default.

## A.3  Evaluation Metrics.

In order to compare the generation performance of existing methods with ours, we adopted six metrics to evaluate the quality, text-image alignment, aesthetics, and human preference of the generated images. They are Frechet Inception Distance (FID) [17], CLIP/BLIP Score [38, 28], Aesthetic Score [44], Human Preference Score (HPS) [49] and ImageRewared [51]. Specifically, FID measures the distribution distances of real and generated image sets utilizing a pre-trained classification network (*e.g.*, Inception V3 trained on ImageNet). Instead, CLIP/BLIP Score calculates cosine similarities between the corresponding image and text features extracted by the CLIP/BLIP image and text encoders, respectively. The Aesthetic Score is based on an additional MLP layer on top of a pretrained CLIP image encoder to measure the human aesthetic aspect of a single image. To consider the alignment with human values and preferences, HPS and ImageReward were proposed and are based on the reward model pretrained on CLIP or BLIP.

However, these metrics do not effectively measure the fidelity of our model to different modalities. Therefore, we additionally employ several other evaluation metrics, including the Learned Perceptual

Table 2: Additional quantitative comparison on the COCO validation set. The best result is highlighted.

**(a) Text + Edge**

| Method | FID ↓ | CLIP Score ↑ | BLIP Score ↑ | Aesthetic Score ↑ | HPS ↑ | ImageReward ↑ |
|---|---|---|---|---|---|---|
| ControlNet | 18.73 | 0.2588 | **0.5338** | **5.39** | 19.53 | **0.3726** |
| T2I-Adapter | 20.21 | **0.2638** | 0.5335 | 5.36 | **19.69** | 0.3344 |
| *Ours* | **16.66** | 0.2561 | 0.5321 | 5.35 | 19.53 | 0.2989 |

**(b) Text + Segmentation Map**

| Method | FID ↓ | CLIP Score ↑ | BLIP Score ↑ | Aesthetic Score ↑ | HPS ↑ | ImageReward ↑ |
|---|---|---|---|---|---|---|
| ControlNet | **20.53** | **0.2640** | **0.5262** | 5.43 | 19.85 | 0.2609 |
| T2I-Adapter | 21.81 | 0.2626 | 0.5225 | 5.25 | 19.57 | 0.0837 |
| *Ours* | 23.88 | 0.2613 | 0.5260 | **5.64** | **19.89** | **0.3735** |

**(c) Text + Pose**

| Method | FID ↓ | CLIP Score ↑ | BLIP Score ↑ | Aesthetic Score ↑ | HPS ↑ | ImageReward ↑ |
|---|---|---|---|---|---|---|
| ControlNet | **35.13** | 0.2656 | 0.5171 | 5.49 | 20.17 | 0.3683 |
| T2I-Adapter | 35.53 | **0.2697** | **0.5211** | 5.45 | 20.19 | 0.3810 |
| *Ours* | 39.64 | 0.2641 | 0.5210 | **5.70** | **20.22** | **0.4488** |

**(d) Text + Sketch + Segmentation + Keypoints**

| Method | FID ↓ | CLIP Score ↑ | BLIP Score ↑ | Aesthetic Score ↑ | HPS ↑ | ImageReward ↑ |
|---|---|---|---|---|---|---|
| *Ours* | 16.70 | 0.2562 | 0.5326 | 5.36 | 19.56 | 0.3118 |

Image Patch Similarity (LPIPS) [54] for overall image quality, the L2 distance for Holistically-Nested Edge Detection (HED), the mean Pixel Accuracy (mPA) [31] and mean Intersection over Union (mIoU) [31] for segmentation maps, and the mean Average Precision (mAP) over 10 object keypoint similarity (OKS) thresholds [26] for human pose map. Specifically, LPIPS is utilized to assess the dissimilarity between the generated image and the ground-truth image. Furthermore, we extract the conditions, namely the sketch map, segmentation map, and pose map, from both the generated image and the ground-truth image. By computing the distance, specifically employing L2 Distance, mPA, mIoU, and mAP, between these extracted conditions, we can gain a deeper understanding of the fidelity with respect to various modalities.

Through these metrics, we can more clearly ascertain whether the generated images can follow the guidance of different modal conditions.

## B  Additional quantitative analysis

We compare Cocktail with two state-of-the-art methods: ControlNet [53] and T2I-Adapter [33], in the context of text-guided image-to-image translation within a single modality. As demonstrated in Table 2, our model not only acquires multi-modal capabilities but also demonstrates performance on par with the comparative methods in single-modal tasks. It is worth noting that FID was measured on the basis of 5000 images from zero-shot generation.

It worth noting that FID may not provide a precise evaluation as they are substantially contingent on the sample size and the characteristics of the training dataset. Our methodology prioritizes the fidelity of the generated images, a perspective that is at odds with the principles of CLIP/BLIP.

## C  Additional qualitative results

We provide additional generated images under various multi-conditional scenarios and compare them with Multi-ControlNet and Multi-Adapter. In Fig. 6, we have demonstrated some samples within less complexity, we further show the comparative performance of our model against other

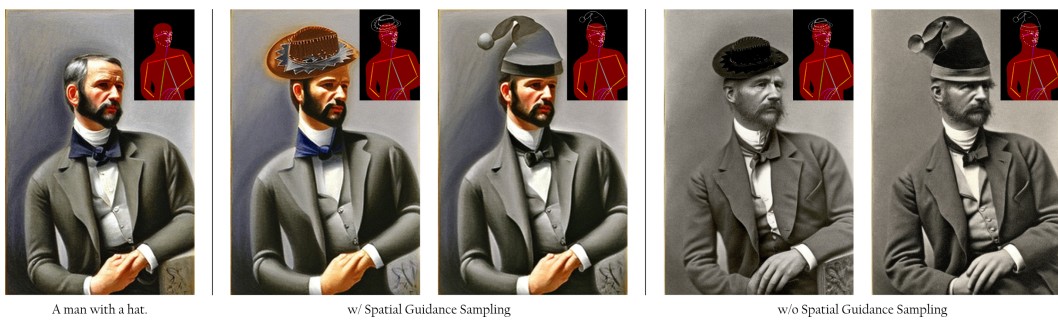

| A man with a hat. | w/ Spatial Guidance Sampling | w/o Spatial Guidance Sampling |

Figure 7: With spatial guidance, our method is capable of maintaining constancy in certain regions while modifying specific objects.

state-of-the-art models under challenging disjoint scenarios in Fig. 8. In Fig. 10, we also conduct an ablation study on the effectiveness of our ControlNorm module. It can be observed that our method outperforms other state-of-the-art models in terms of balancing and expressing multi-modal signals, and ControlNorm provide a more stable integration performance. Besides, our model is able to accept arbitraty combiantions of the given modalities, as shown in Fig. 9.

We also provide further samples under single modality signals, *e.g.*, Human Pose in Figs. 11&12, Sketch Map as shown in Figs. 13&14 and Segmentation Map in Fig. 15. It can be seen that our method is more faithful to the given conditional signals.

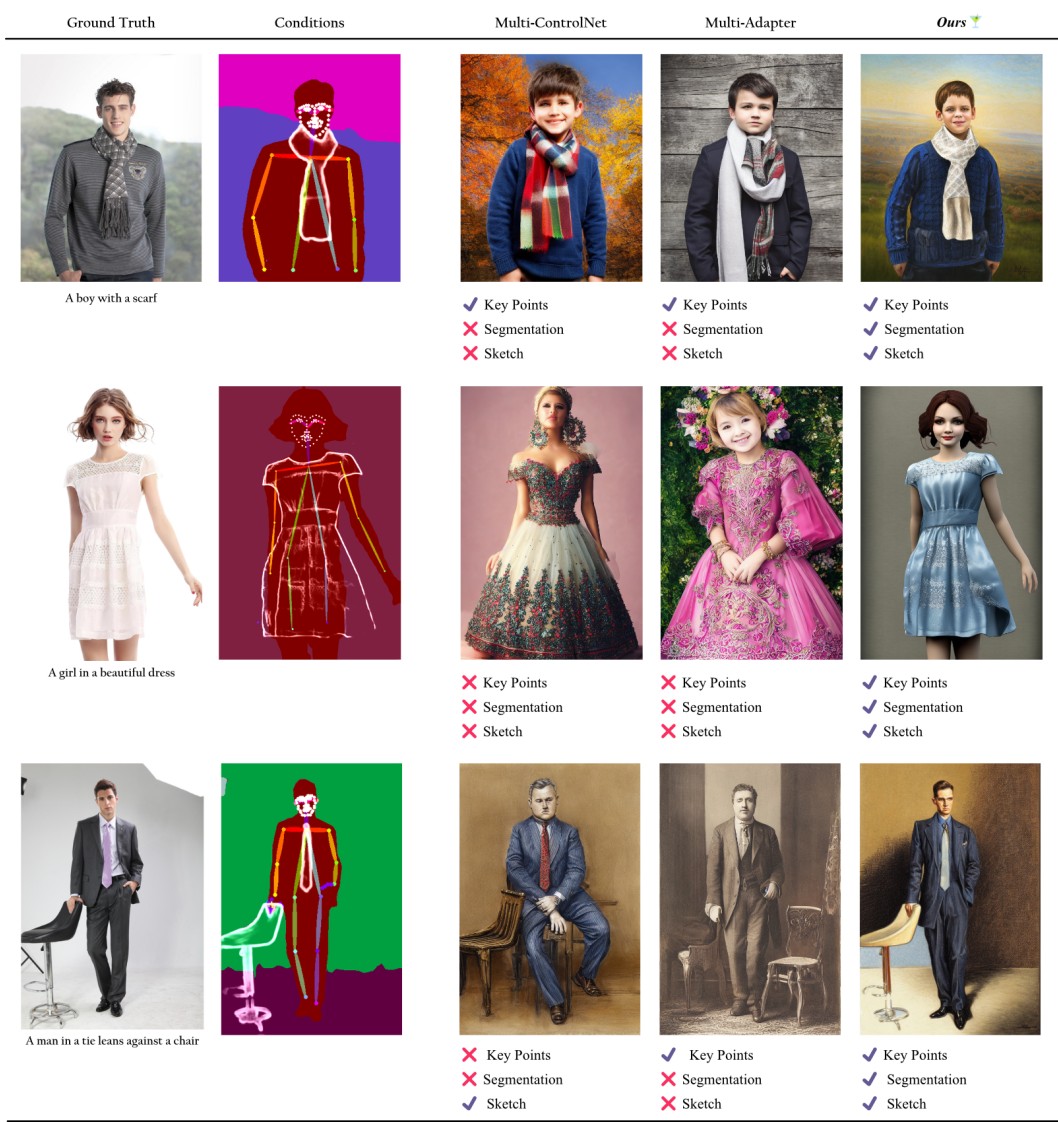

Figure 8: Additional results for Disjoint Multi-Modality Control.

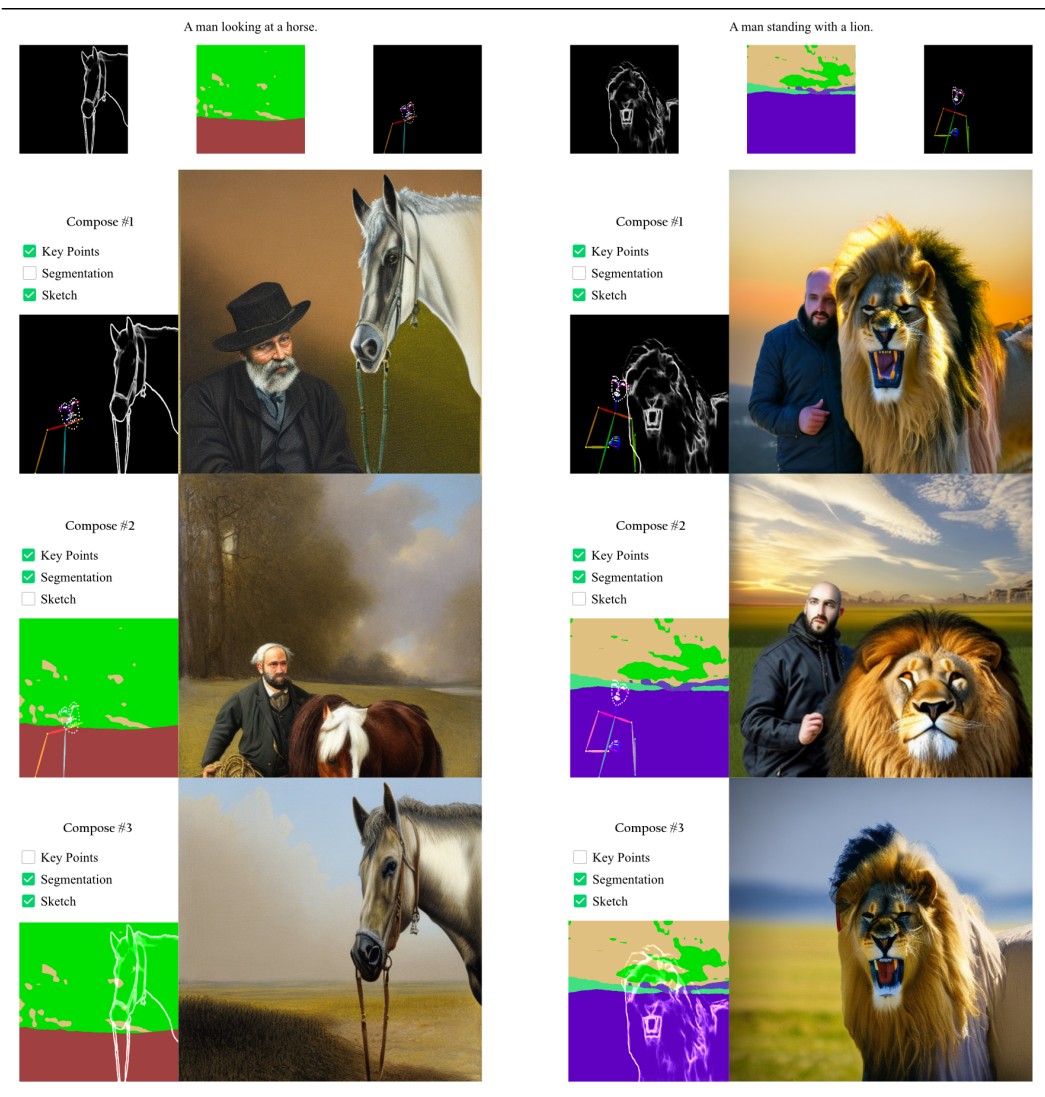

Figure 9: Our model accepts arbitrary combinations of the given modalities.

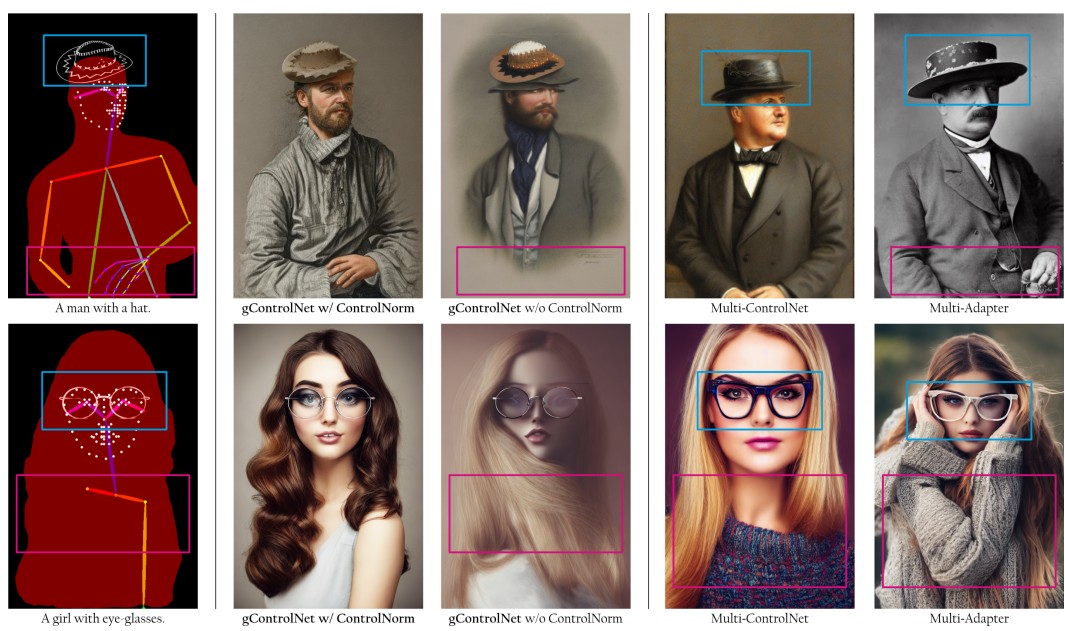

Figure 10: ControlNorm can address the imbalance among various modalities. Note that the framed areas in magenta and cyan do not provide a high-fidelity interpretation.

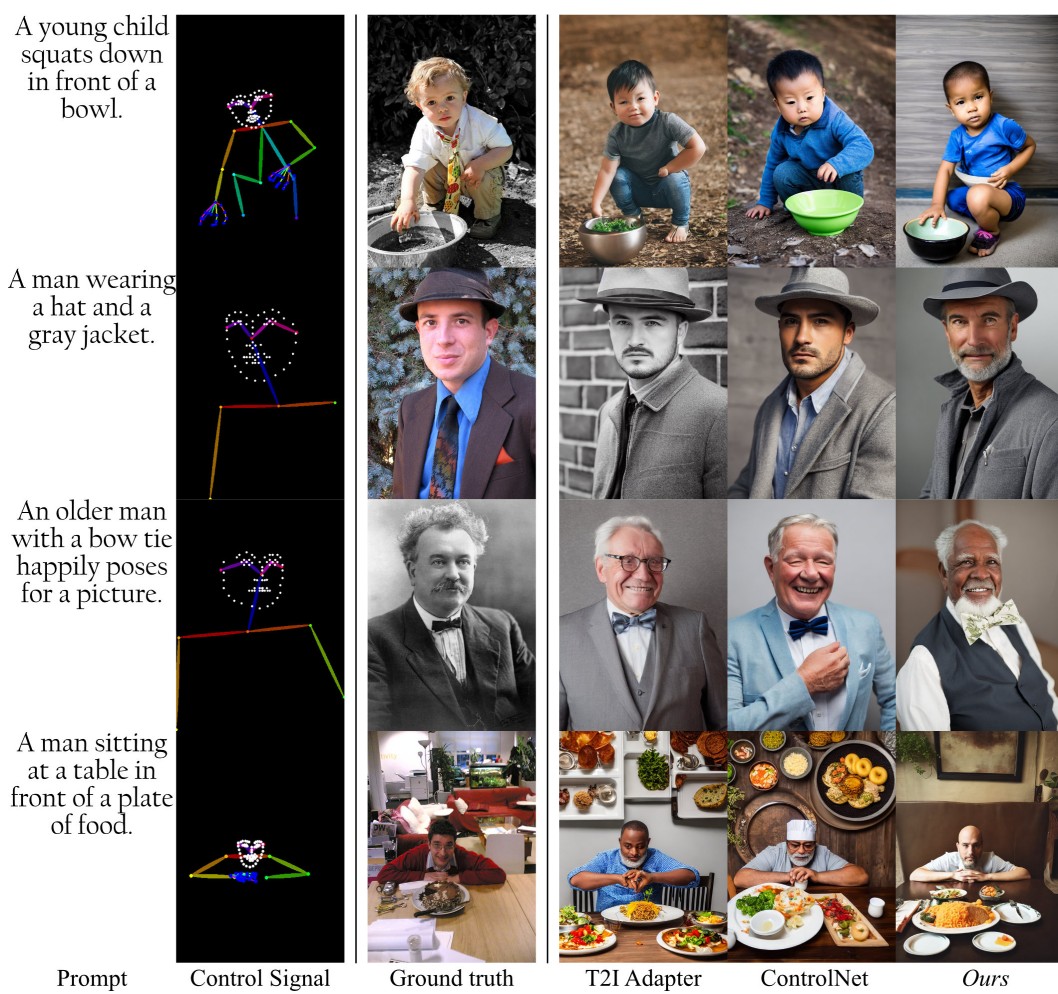

Figure 11: Additional results for Single Modality Control: Key Points

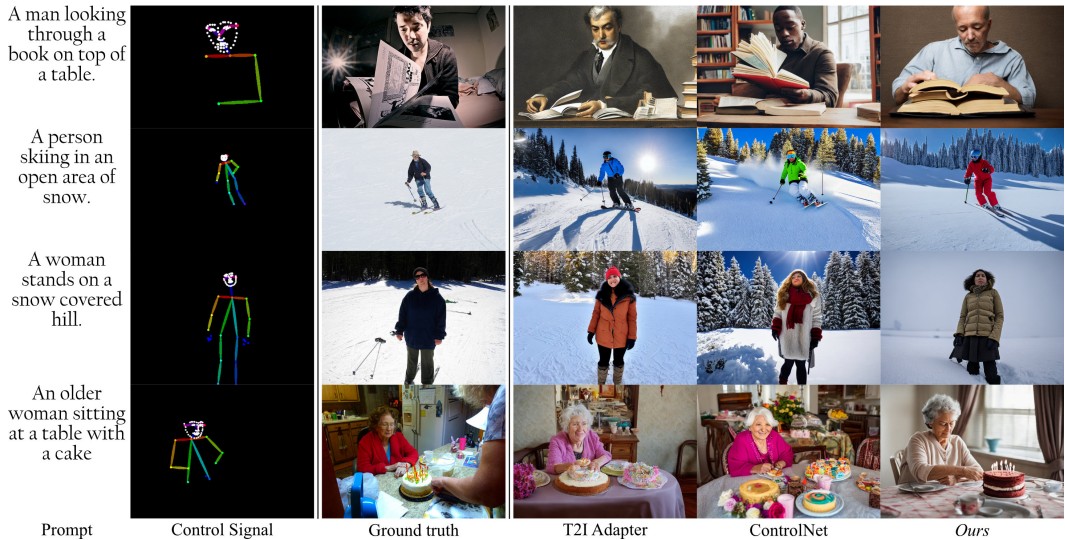

Figure 12: Additional results for Single Modality Control: Key Points

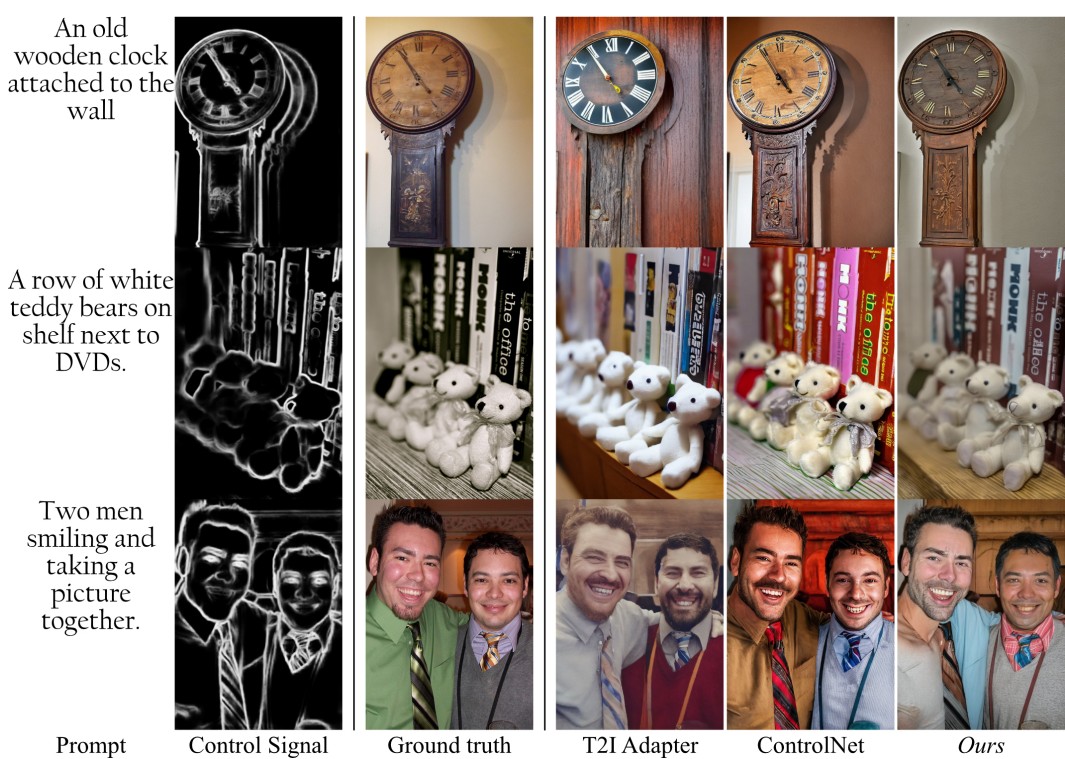

Figure 13: Additional results for Single Modality Control: Sketch

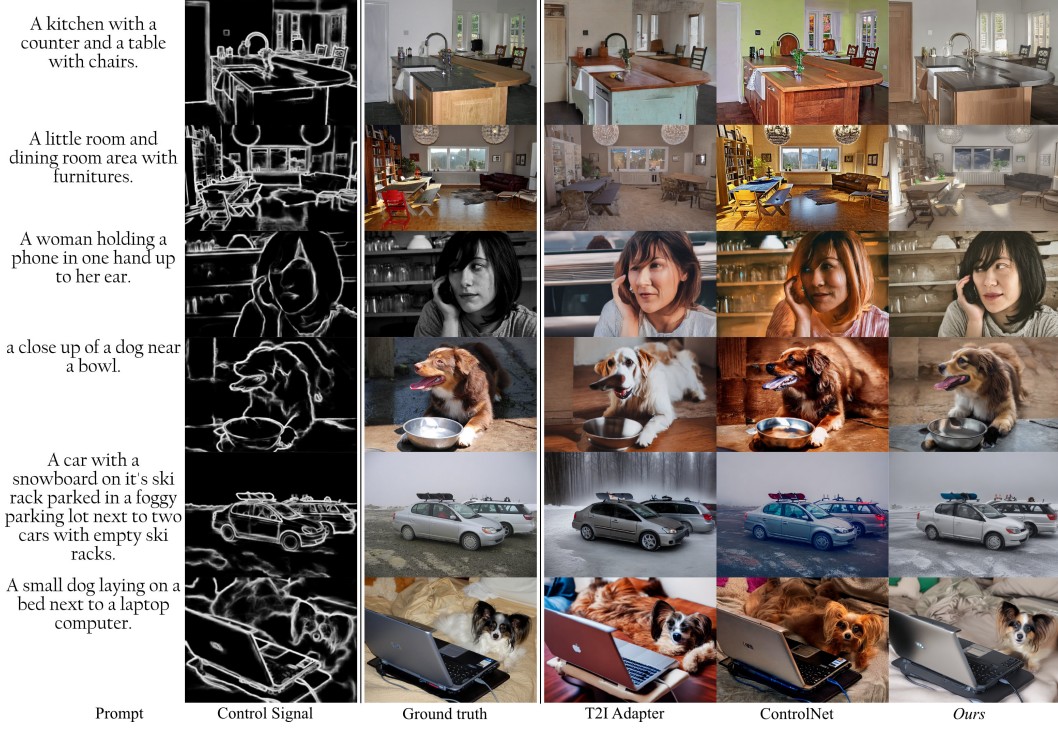

Figure 14: Additional results for Single Modality Control: Sketch

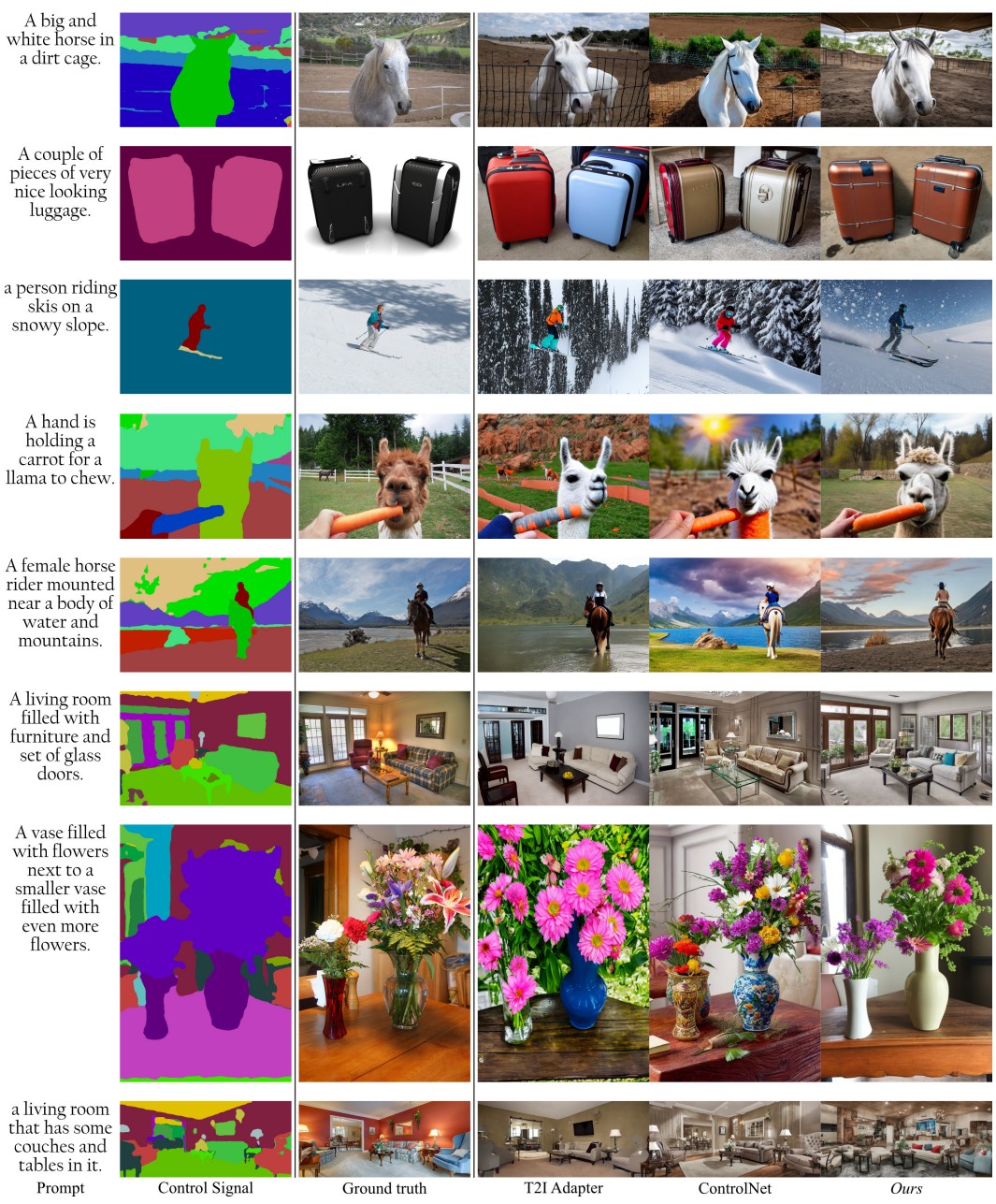

| Prompt | Control Signal | Ground truth | T2I Adapter | ControlNet | *Ours* |

Figure 15: Additional results for Single Modality Control: Segmentation Map

