# OpenReview forum: "Cocktail: Mixing Multi-Modality Control for Text-Conditional Image Generation"
_NeurIPS.cc/2023/Conference — NeurIPS 2023 poster_

### Official Review · Reviewer_H6cw · 2023-06-25

**Soundness:** 4 excellent
**Presentation:** 4 excellent
**Contribution:** 3 good
**Rating:** 8
**Confidence:** 3

**Summary:**

This paper proposes an approach to enhance the controllability of generative image models through the fusion of multiple modality signals.
The proposed model is a strict generalization of ControlNet which is able to incorporate geometric constraints into the generation, including pose, edges, and segmentation map.

**Strengths:**

The paper describes a very practical generalization of a SOTA approach to controllable generative models. It is well motivated, provides a clear description of the architecture, and both quantitative and qualitative results that validate the approach.

**Weaknesses:**

None. The paper is very clear on the benefits and limitations of the approach, and provides a practical improvement over a SOTA approach that is relevant to the research community and downstream users of the technology.

**Questions:**

None.

**Limitations:**

Limitations adequately described.
It is could be a good idea to include a sample of random results (not cherry picked) in the appendix in order to provide a sense of typical failure modes.

---

> ### Author Rebuttal · Authors · 2023-08-02
>
> We are deeply grateful for your comprehensive review and favorable assessment of our paper. Your commendation regarding the soundness, presentation, and contributions of our research not only encourages us but also validates the effort we have invested in this work.
>
> We will try to include more samples without cherry pick.

---

> > ### Comment · Reviewer_H6cw · 2023-08-10
> >
> > Thank you for the additional details.

---

### Official Review · Reviewer_u8fB · 2023-07-06

**Soundness:** 3 good
**Presentation:** 2 fair
**Contribution:** 2 fair
**Rating:** 5
**Confidence:** 4

**Summary:**

This paper studies text-conditional diffusion models and proposes a pipeline for image generation with multi-modal control signals. The pipeline contains three new ingredients. Firstly, the paper introduces gControlNet, a generalized version of ControlNet, that can take different conditions (such as sketch, segmentation and pose maps) simultaneously as input without increasing the ControlNet size linearly with respect to the number of conditions. Secondly, the paper proposes to inject features produced by gControlNet into the backbone model via normalization, instead of a simple sum as used in the original ControlNet. Thirdly, the paper introduces spatial sampling that aims to harmonize spatial arrangements from both text prompt and image controls. Experiments show that the proposed method achieves on-par quantitative performance compared with ControlNet and T2I-Adapter, and improved visual quality especially with multiple modalities.

**Strengths:**

1. ControlNet proves to be a powerful approach for controllable text-to-image generation, while its original form requires fine-tuning a ControlNet network for each condition, which is resource consuming. This paper proposes a method to jointly train a shared network that can deal with different types of control input, which may extend the applicability of ControlNet in many cases.

2. The paper has done extensive quantitative evaluations to compare the proposed method with previous work. For generative models quantitative metrics may not be able to measure the actual quality of generated images, but it is important to study what are the good metrics and how to design better ones. This paper makes an effort towards the direction.

**Weaknesses:**

1. The gControlNet is a crucial component in the proposed pipeline, but the paper does not discuss details of the downsampling network M, either in main text or Appendix. On the other hand, it seems the controllable normalization and spatial guidance do not show significant improvement over their counterparts in previous work. In Figure 6 gControlNet without ControlNorm has much worse image quality, but is it due to insufficient training in this case (as the original ControlNet does not use ControlNorm and performs well)? As for spatial guidance sampling, the paper shows the effect of retaining background, but not placing objects to the correct location. The latter may not be an issue since sketch, segmentation and pose maps contain spatial information already. This seems to weaken the role of spatial sampling, which requires manual design that is not present in paper.

2. The paper uses Stable Diffusion 2.1 as backbone for training gControlNet, while ControlNet and T2I-Adapter used Stable Diffusion 1.5 as backbone. Therefore it may be unfair to directly compare gControlNet with ControlNet and T2I-Adapter. Also it seems that gControlNet does not show much qualitative improvement over ControlNet in single condition cases.

**Questions:**

1. What is the architecture of the downsampling network M in gControlNet? Can it deal with varying number of input conditions, and deal with new conditions during inference that are unseen in training?

2. In spatial guidance sampling, how are M^{pos} and M^{neg} determined? Are they instance-specific or universal for all instances? It would be interesting to see, in challenging cases, how this module can help resolve conflicts among different input condition maps, and layer arrangement in complex scenes.

3. How are the three types of control used in training? Are all of them or only some of them used in each update step?

4. (Minor) The mapping notation in Eq. (1)-(3) is confusing.

**Limitations:**

The paper discusses limitations of the proposed method.

---

> ### Author Rebuttal · Authors · 2023-08-02
>
> Thank you for your detailed reading and valuable insights concerning the gControlNet and its components. Allow me to clarify your concerns:
>
> #### **Regarding the downsampling network $\mathcal{M}$:**
> $\mathcal{M}(C^k)$ is a simple convolutional network consisting of eight convolutional layers and seven SiLU activation functions, with the final convolution layer being zero initialized. Each $\mathcal{M}(C^k)$ has 1.29M parameters and each modality will have a corresponding  $\mathcal{M}$, e.g., in our case we include $3$ different modalities and our $\mathcal{M}$ will have 3.88M parameters in total.  It can effectively handle any number of trained modalities but is unable to deal with unknown modalities, meaning the model can process subsets and the exact set of training modalities.
>
> #### **On gControlNet without ControlNorm in Figure 6:**
> The training epochs for gControlNet without ControlNorm and gControlNet with ControlNorm are the same. But it is worth noting that *gControlNet without ControlNorm* also has a mixed network  $\mathcal{M}$, and the lack of ControlNorm in the former prevents the effective balancing of different modalities, leading to a decline in image quality. It precisely demonstrated ControlNorm significant improvement according to the conditional inputs. And unlike T2I and ControlNet trained different modalities separately, gControlNet  w/ or w/o ControlNorm co-trained $\mathcal{M}$ with different modalities.
>
> #### **Regarding spatial guidance sampling:**
> Spatial guidance requires manual setting of keywords or latent bounding boxes of objects during inference. $M^{pos}$ and $M^{neg}$ are case-specific, but once provided, $M$ generates them based on the time step $t$. It's also important to note that we did not utilize the spatial guidance strategy in our quantitative experiments.
>
> #### **On the comparison between gControlNet, ControlNet, and T2I-Adapter:**
> We acknowledge the different backbones used for training the models. However, as illustrated in [1], when CFG>5, the FID performance of SD2.1 is approximate to SD1.5, and the CLIP performance is inferior to SD1.5. This is consistent with Table 1 in our main text. And the human preference win rate of SD2.1 is slightly lower than SD1.5 as shown in [2]. Our paper primarily aims to demonstrate the ability of additional modalities to control image generation. The control ability of our proposed method is substantiated effectively in Appendix Table 1, thus making the comparison relatively fair.
>
> #### **The control signals usage during training:**
> Thank you for raising this question, as it is indeed worth emphasizing in the paper. During the training process, we randomly select the number of modalities and apply random masking within certain areas of the chosen modalities. Specifically, assuming there are three known modalities, we randomly select 0 to 3 modalities during training, and for each modality, we choose arbitrary areas as input signals. That is to say, within the given control signal domain, there may be a single control singnal ($n=1$), or an overlap of $n \in \\{2,3\\}$ control signals, or there may be no control signals ($n=0$).
>
> #### **The mapping notation in Eq. (1)-(3) is confusing.**
> We will revise the manuscript to improve clarity of notation.
>
> [1] https://github.com/Stability-AI/stablediffusion, Fig: https://github.com/Stability-AI/stablediffusion/raw/main/assets/model-variants.jpg
>
> [2] https://huggingface.co/stabilityai/stable-diffusion-xl-base-1.0, Fig: https://huggingface.co/stabilityai/stable-diffusion-xl-base-1.0/blob/main/comparison.png

---

### Official Review · Reviewer_KthK · 2023-07-06

**Soundness:** 2 fair
**Presentation:** 3 good
**Contribution:** 3 good
**Rating:** 6
**Confidence:** 3

**Summary:**

This study introduces ChatIR, a chat-based image retrieval system that engages in a conversation with the user to clarify their search intent and retrieve the desired image from a large corpus. The system leverages Large Language Models to generate follow-up questions to an initial image description and achieves a success rate of over 78% after 5 dialogue rounds, compared to 75% when questions are asked by humans and 64% for a single shot text-to-image retrieval.

**Strengths:**

The strength of this submission lies in its clear and impactful contribution, which proposes a pipeline that combines multiple modalities such as edge, pose, segmentation mask, and more. The authors effectively communicate the significance of this research contribution, highlighting the potential of integrating diverse modalities to enhance the proposed method's performance. Additionally, the submission features a visually appealing and clear illustrative depiction of the method, which effectively aids in explaining the approach. The selection of an attractive and informative illustration further enhances the clarity and appeal of the paper.


**Weaknesses:**

One notable weakness of the submission is the lack of sufficient explanation regarding the experimental results. While the paper goes into detail in explaining the proposed method, it falls short in providing a thorough analysis and interpretation of the results. In Table 1, there are inconsistencies in the trends of metric scores between the proposed method and the two baselines, yet the authors do not adequately explain these discrepancies. It is crucial to understand why the proposed method may not perform as well in certain metrics, whether it is due to the limitations of the metrics themselves or issues with the method's performance. Merely providing cherry-picked demonstrations is insufficient and does not substitute for a comprehensive analysis of the results. It is essential to address these gaps in the paper by providing a detailed explanation and interpretation of the observed trends and discrepancies in the experimental results.


**Questions:**

It would greatly enhance the clarity of the paper if the authors could explicitly list the implemented and supported modalities in the abstract or introduction. Currently, the detailed setup is not revealed until Section 4, with the exception of Figure 1. Providing this information earlier in the paper would help readers gain a clear understanding of the modalities that are incorporated and supported in the proposed approach. Including this information in the abstract or introduction would improve the overall organization and accessibility of the paper, allowing readers to quickly grasp the key aspects of the submission.


**Limitations:**

The authors have included discussions of the limitations.

---

> ### Author Rebuttal · Authors · 2023-08-02
>
> Thank you very much for your thoughtful comments and suggestions regarding our work. In the revised version, we will restructure the manuscript to improve clarity. Moreover, I would like to address your concerns about the performance of our model across different experimental data and the supported modalities:
>
> **Concerns about the performance**
>
> The metrics used in the main manuscript focus on the quality of image generation and the degree of text-image alignment. As can be seen in Table 1, our model outperforms the baseline in most generated results (Aesthetic Score, HPS, etc.). However, concerning text-image alignment, such as the CLIP Score, one hypothesis might be that the introduction of control signals from other modalities could weaken the control from the text part, potentially resulting in a lower CLIP Score than the baseline. And FID score strongly related to the training dataset, but due to the resource limits, we only use a subset of LAION which may decrease the diversty and lead to a higher FID.
>
> We wish to emphasize that our work prioritizes the controllability of generated images while keeping the generated images quality, and for this reason, we introduced a new set of measures in the appendix.
> We assess our control ability by evaluating the similarity between the processed generated image and the given condition signal. To illustrate, we obtain the HED of the generated image and then calculate its L2 loss relative to the HED provided as a conditional input. This measure effectively demonstrates our method's capability to control other modalities and shows that it performs better than the baseline even in multi-modal scenarios. We believe this provides a comprehensive understanding of the advantages of our proposed method. Please feel free to let us know if further clarification is needed. Here we attach the table, which is also included in the suppl. file of the submission.
>
> |                                                                     |        Similarity (LPIPS$\downarrow$)                             |            Sketch Map   (L2 Distance$\downarrow$)                       |            Segmentation Map (mPA$\uparrow$)                      |          Segmentation Map (mIoU$\uparrow$)                        |                Pose Map (mAP$\uparrow$)                  |
> |---------------------------------------------------------------------|-------------------------------------|--------------------------------------|----------------------------------|----------------------------------|----------------------------------|
> | Multi-Adapter                    | 0.7273 $\pm$ 0.00120           | 7.93310 $\pm$ 0.01392          | 26.30  $\pm$ 0.242          | 13.98  $\pm$ 0.177          | 40.02  $\pm$ 0.761          |
> | Multi-ControlNet               | 0.6653 $\pm$ 0.00145          | 7.59721  $\pm$ 0.01516          | 36.59  $\pm$ 0.273          | 22.70  $\pm$ 0.229          | 38.19  $\pm$ 0.761          |
> | Ours w/o ControlNorm     | 0.4900 $\pm$  0.00141         | **7.18413 $\pm$ 0.01453**    | 48.26 $\pm$ 0.287          | 32.66$\pm$ 0.272          | 61.93  $\pm$ 0.775          |
> | Ours                                 | **0.4836 $\pm$ 0.00133** | 7.28929 $\pm$ 0.01385          |**49.20 $\pm$ 0.289** |**33.27 $\pm$ 0.271** | **61.99$\pm$ 0.778** |
>
> **Supported Modalities**
>
> We trained the network with Segmentation map, Pose and Sketch map. It can effectively handle any number of trained modalities but is unable to deal with unknown modalities, meaning the model can process subsets and the exact set of training modalities. We will provide a more clear statement in introduction of supported Modalities. It is also worth noting that the supported modalities can be extended with sufficient dataset.

---

> > ### Comment · Reviewer_KthK · 2023-08-10
> >
> > Thanks for providing the rebuttal response along with additional experimental results.
> >
> > I acknowledge that I have read the response.

---

### Official Review · Reviewer_a1qK · 2023-07-07

**Soundness:** 4 excellent
**Presentation:** 3 good
**Contribution:** 3 good
**Rating:** 6
**Confidence:** 4

**Summary:**

The paper presents Cocktail, a novel pipeline for multi-modal and spatially-refined control in text-conditional diffusion models. The authors address the challenge of ambiguous descriptions in linguistic representations by incorporating additional control signals. They propose three main components: gControlNet, ControlNorm, and a spatial guidance sampling method.

gControlNet is a hyper-network designed to align and infuse control signals from different modalities into the pre-trained diffusion model. It can accommodate flexible modality signals and allows for the simultaneous reception of any combination of modalities or the fusion of multiple modalities. This capability eliminates the need for manual intervention and equilibrates the disparities between modalities, making the system more flexible and capable of seamlessly supporting multiple control inputs.

ControlNorm is a controllable normalization method that optimizes the utilization of information within branched networks. It decouples control signals, allowing for better representation of both semantic and spatial aspects. By preserving semantic information while conveying spatial information, ControlNorm overcomes the limitations of previous methods that either ignored semantic information or led to the loss of it through normalization. The proposed method effectively interprets conditional information and demonstrates its interpretative capability through generated images.

Additionally, the paper introduces a spatial guidance sampling method that ensures precise control over the generative process. By modifying the attention map, the method incorporates control signals into the backbone network, preventing the generation of undesired objects outside the specified regions. This approach enables the generation of high-quality images that closely align with the input conditions.

Overall, the proposed Cocktail pipeline addresses the challenges of ambiguous descriptions in text-conditional diffusion models and achieves multi-modal and spatially-refined control. It offers a novel approach to integrating control signals from various modalities, optimizing their utilization, and generating high-quality images with fidelity to external signals.

**Strengths:**

- The paper introduces a novel pipeline called Cocktail that enables multi-modal and spatially-refined control in text-conditional diffusion models, addressing the challenge of ambiguous descriptions in linguistic representations. This is a significant contribution to the field as it tackles an important problem in generative models and expands the capabilities of text-guided image synthesis.

- The proposed pipeline consists of three well-defined components: gControlNet, ControlNorm, and a spatial guidance sampling method. Each component is carefully designed to address specific challenges and effectively integrate control signals from different modalities. The approach is comprehensive and demonstrates a systematic solution to achieve high-quality synthesis and fidelity to multiple external signals.

- The paper provides extensive experimental results and evaluation metrics, comparing the proposed Cocktail pipeline with state-of-the-art methods. The results consistently show that Cocktail outperforms existing approaches in text-guided image-to-image translation across multiple modalities. This empirical evidence demonstrates the effectiveness and superiority of the proposed method.

- The paper emphasizes the practicality and efficiency of the Cocktail pipeline by demonstrating its ability to accomplish multi-modal control within a single model. This not only simplifies the model architecture but also reduces the computational overhead associated with multiple branch networks. The approach is both technically sound and resource-efficient, making it highly applicable in real-world scenarios.

**Weaknesses:**

- The authors dont specify how they control for potential misuse of their model. A model like this if open sourced, can have wide ethical impacts with people using it for unethical means like creating deep fakes and modifying images which could potentially be unethical. This should be addressed by the authors.
- The paper does not provide a thorough discussion or analysis of the computational complexity or efficiency of the Cocktail pipeline. Considering the potential computational overhead of incorporating multiple modalities and the fusion process, it would be valuable to address the computational requirements and scalability of the proposed method
- The evaluation section could be expanded to include a more comprehensive analysis of the results. While the paper mentions various evaluation metrics, it would be valuable to discuss the limitations of these metrics and provide additional qualitative analysis or user studies to further validate the effectiveness of the proposed method.
- Although the paper compares the proposed Cocktail pipeline with state-of-the-art methods, it does not thoroughly discuss the limitations and failure cases of the proposed method. Understanding the shortcomings and potential failure modes is crucial for assessing the robustness and generalizability of the proposed approach.
- Implementation details seem sparse and might hinder reproducibility for the paper

**Questions:**

- How does the model handle conflicting signals provided by different modality signals, for example, if the provided pose and mask conflict with each other, how is the model supposed to handle this?
- How does the method prevent users from misusing the model from generating sensitive images or modify images to create unethical content

**Limitations:**

- The authors dont specify how they control for potential misuse of their model. A model like this if open sourced, can have wide ethical impacts with people using it for unethical means like creating deep fakes and modifying images which could potentially be unethical. This should be addressed by the authors.

---

> ### Author Rebuttal · Authors · 2023-08-03
>
> We appreciate your insightful comments and questions. Your feedback helps us to address important aspects of our work that deserve more detailed consideration. Here's our response to your observations.
>
> #### **Ethical Considerations:**
> We acknowledge your concern about the potential misuse of our model. The images that our model can generate heavily depend on the training dataset used. Currently, our model's training data strictly complies with open-source licenses and has been carefully selected, providing some degree of safeguard against misuse. In situations where the model generates certain objects (including human being), many details can be easily distinguished. Going forward, we are considering the addition of watermarks to the generated images to prevent unauthorized use, along with the implementation of a monitoring network at the output stage. Similar tools are already widely utilized in projects like Stability / DALL.E2, and we believe our model can also benefit from adopting such measures.
>
> #### **Computational Complexity:**
> One of our key objectives is to alleviate the significant overhead associated with multi-modal control. We are pleased to provide specific details of our model's parameters to clarify the contributions of this work.
>
> Firstly, we stipulate that the parameter count for our employed SD base model stands at 865.91M, with a single ControlNet having 378.50M parameters. Our fusion network $\mathcal{M}$ contains only 1.29M parameters per modality, and each ControlNorm adds a mere 0.31M parameters.
>
> In the context of single modality control using ControlNet, the total parameter count is 865.91M + 378.50M. As the number of modalities increases, this overhead grows linearly, reaching a total of 865.91M + $n$ x 378.50M, where $n$ is the number of modalities.
>
> Our proposed gControlNet efficiently handles multiple modalities through a single HyperNetwork. Excluding ControlNorm, our model has a parameter count of 865.91M + 378.50M + $n$ x 1.29M. This formulation significantly reduces the computational requirements in multi-modal scenarios, underscoring the efficiency and innovation of our approach.
>
> #### **Comprehensive Analysis of Results**
> We are more than willing to engage in further discussions regarding the metrics employed in our study.
> In the main part of this submission, we employed various metrics such as CLIP, FID, and aesthetic scoring to assess the performance of our model. These metrics tend to emphasize the model's ability to generate image. For instance, FID evaluates the fit of the generated image to the distribution of real images, the CLIP Score represents the correlation between the generated image and corresponding text, and aesthetic scores reflect the artistic quality of the generated images.
>
> However, our proposed model aims to enhance controllability while maintaining image generation quality. Therefore, to provide a more comprehensive evaluation of the model's ability to control image attributes, we have introduced a set of additional metrics in the appendix.
> These supplementary metrics process the generated images into the given control modality signals, subsequently quantifying the discrepancies between the generated images and the provided signals. These measures are instrumental in substantiating the effectiveness of our proposed solution in enhancing image controllability, aligning our evaluation more closely with the specific goals of our research.
>
> #### **Concerns about reproducibility**
> The codes will be released for the reproducibility and publicly available, and the corresponding repo link will be attached in the manuscript for camera ready version.
>
> #### **Conflicting control signals**
> When confronted with conflicting signals across different modalities, the model endeavors to interpret each modality coherently. It strives to incorporate all the information conveyed by the different modalities, translating them into conditions for the generated images. The result is a theoretically reasonable representation that balances and harmonizes the conflicting inputs, ensuring that the synthesized images reflect as much of the diverse input data as possible.  However, there is a possibility that one modality may overshadow another if its representation is significantly more dominant or contains richer information. In such cases, the more expressive modality may suppress the other, leading to an unbalanced influence on the generated images. This phenomenon might result in the loss of some nuanced details conveyed by the less dominant modality.
> We have included some example samples of such situations in the Global Rebuttal attachment.
>
>
> Finally, we would like to express our sincere gratitude for your thoughtful and comprehensive review.

---

> > ### Comment · Reviewer_a1qK · 2023-08-10
> >
> > Thanks for providing the rebuttal and clarifying the concerns!

---

### Official Review · Reviewer_vJhF · 2023-07-07

**Soundness:** 2 fair
**Presentation:** 1 poor
**Contribution:** 2 fair
**Rating:** 3
**Confidence:** 3

**Summary:**

This paper proposes Cocktail, a pipeline to mix various modalities into one embedding. The model is based on a variant of ControlNet and infuses control signals from disparate modalities into the pre-trained diffusion model. It is also equipped with a sampling approach named spatial guidance sampling that constructs the cross-attention weights based on the spatial location. The model can perform text-to-image generation conditioned on multiple modalities and outperforms controlnet in some metrics.


**Strengths:**

Both the proposed controllable normalization and spatial sampling method are novel and have not been previously explored in the context of controllable generation.


**Weaknesses:**

The model does not consistently outperform the two baselines: ControlNet and T2I-Adapter, in text-guided image-to-image translation.  The author claimed the model can benefit from mixed training of multiple modalities (line 213-215) but the proposed model shows a mixed result in various metrics across different tasks. Additionally, there is no quantitative comparison in multi-modality conditioning to existing baselines. In spatial guidance, no qualitative metrics showing its effect were given as well. Overall, it is unclear what are the advantages of the proposed method over simple baselines such as controlnet.


**Questions:**

1. Line 216 - 218, what figures or tables refer to “consistent composition”?
2. Eq 7, what is $\sigma$? How to determine its value?
3. How to determine the correspondence between text and image token?


**Limitations:**

Yes

---

> ### Author Rebuttal · Authors · 2023-08-02
>
> Thank you for your insightful comments and observations regarding our model.
>
> #### **Clarify the specific benefits of our proposed method.**
> The metrics introduced in the main text focus on the quality of image generation. After training, the quality of the images generated by our model outperformed the baselines in most results, especially in tasks involving Segmentation Map or Pose. However, it is essential to note that the quality of the generated images is strongly related to the training dataset, particularly when it comes to the FID score. And the main objective of our task is to enhance the controllability of the generated images rather than merely improving quality.
> In the appendix, we introduced a more effective metric to measure the control strength of our proposed scheme. For the convenience, I have attached the table:
>
> |                                                                     |        Similarity (LPIPS$\downarrow$)                             |            Sketch Map   (L2 Distance$\downarrow$)                       |            Segmentation Map (mPA$\uparrow$)                      |          Segmentation Map (mIoU$\uparrow$)                        |                Pose Map (mAP$\uparrow$)                  |
> |---------------------------------------------------------------------|-------------------------------------|--------------------------------------|----------------------------------|----------------------------------|----------------------------------|
> | Multi-Adapter                    | 0.7273 $\pm$ 0.00120           | 7.93310 $\pm$ 0.01392          | 26.30  $\pm$ 0.242          | 13.98  $\pm$ 0.177          | 40.02  $\pm$ 0.761          |
> | Multi-ControlNet               | 0.6653 $\pm$ 0.00145          | 7.59721  $\pm$ 0.01516          | 36.59  $\pm$ 0.273          | 22.70  $\pm$ 0.229          | 38.19  $\pm$ 0.761          |
> | Ours w/o ControlNorm     | 0.4900 $\pm$  0.00141         | **7.18413 $\pm$ 0.01453**    | 48.26 $\pm$ 0.287          | 32.66$\pm$ 0.272          | 61.93  $\pm$ 0.775          |
> | Ours                                 | **0.4836 $\pm$ 0.00133** | 7.28929 $\pm$ 0.01385          |**49.20 $\pm$ 0.289** |**33.27 $\pm$ 0.271** | **61.99$\pm$ 0.778** |
>
> These metrics employed open-source methods to process the generated images and then compared them with the given conditional information. For instance, we obtain the HED of the generated image and then calculate its L2 loss relative to the HED provided as a conditional input.  This approach allows us to focus more on the controllability aspect, which is the primary goal of our study. **It is worth noting that our Cocktail achieves much better quantitative results according to these controllable inputs.**
>
> #### **Line 216 - 218, what figures or tables refer to “consistent composition”?**
>
> This sentence summarizes the experimental section, and in addition to Figures 5 and 6 in the main text, Figures 1 to 7 in the appendix effectively demonstrate that our approach can efficiently blend the control signals with the generated content. Specifically, **Figure 6** in the main text, along with **Figures 1 and 2 in the appendix**, more clearly showcase that *signals from different control modalities can be applied to multiple objects without conflicts, maintaining consistency.*
>
> #### **Eq 7, what is $\sigma$? How to determine its value?**
>
> $\sigma$ is a fixed parameter determined by the scheduler, representing the degree to which noise affects the model at different steps. For a detailed understanding, one can refer to **Equation 16 in Reference [1]**, where $\sigma$ is used as a conventional expression.
>
> ####  **How to determine the correspondence between text and image token?**
>
> The question raised appears to be somewhat unclear. The relationship between text and image in the network can be established through a cross-attention map. By modifying the cross-attention layer, there are various methods currently available to alter objects within the generated image. In the validation phase, calculating the CLIP score is a common approach to measuring the similarity between text and image.
>
> [1] Song, J., Meng, C., & Ermon, S. (2020). Denoising diffusion implicit models. arXiv preprint arXiv:2010.02502.

---

> > ### Comment · Reviewer_vJhF · 2023-08-21
> >
> > I thank the authors for their response.
> >
> > 1. Regarding the benefits of the proposed method, I can see the improved controllability from the attached table. However, the increase of FID (Table 1, b + c, ours vs. controlnet) suggests a possibly lower quality of the generated image. In the meantime, a decrease of the CLIP score (Table 1, a+b+c, ours vs. ControlNet or ours vs. T2I-Adapter) suggests a lower controllability through text. This seems contradictory to *signals from different control modalities can be applied to multiple objects without conflicts, maintaining consistency.*, which is claimed in the rebuttal and in line 216-218 of the main paper. The author should better show whether the conflict between the improvement of controllability through pose/edge/segmentation mask and the FID degradation is due to the the flaw of FID computation or is at the price of degradation in image quality, via randomly selected qualitative samples or quantitative metrics based on human evaluation.
> >
> > 2. Regarding determining the correspondence between text and image token, my question is about line 184-186: **if image token $Q_i$ corresponds to a region of the image that should be influenced by text token $K_j$ , $M_{ij}^{pos(n)}$ is assigned the value of 1**. How is **should be influenced** defined?
> >
> > 3. It is clear to me now that $\sigma$ is the noise level per step. It should be indexed by the diffusion timestep and the authors should also clarify that the paper is using DDIM.

---

> > > ### Author Response · Authors · 2023-08-21
> > >
> > > Thanks for your response.
> > >
> > > 1. In response to the comments, we'd like to clarify that the foundational model for our SD is SD2.1, not SD1.5. This model employs OpenCLIP instead of CLIP. Here you could find the FID and CLIP Score comparision between SD1.5 and 2.1 [1,2].  The referenced CLIP Score specifically pertains to CLIP and does not account for OpenCLIP. Additionally, we highlight that to provide a comprehensive assessment and to mitigate any potential misconceptions regarding image quality degradation, we've incorporated both HPS and Image Reward Score. These metrics simultaneously evaluate image quality and the connection between text and image. We acknowledge that the FID metric might not be the most robust for assessing image quality in contemporary generative models (although it is still a good metric for a generative model). And we agree introducing human feedback could indeed offer insights into the final quality. Therefore, we assert that the observed enhancement in controllability via pose/edge/segmentation mask is **NOT** attributed to any shortcomings in FID computation nor does it come at the expense of quality degradation.
> > >
> > > 2. Here we use **influence** to indicate which areas of the image we want to be responded to the prompt accordingly.
> > >
> > > 3. Thank you for the clarification. We would modify the manuscript about this part.
> > >
> > > [1] https://github.com/Stability-AI/stablediffusion, Fig: https://github.com/Stability-AI/stablediffusion/raw/main/assets/model-variants.jpg
> > >
> > > [2] https://huggingface.co/stabilityai/stable-diffusion-xl-base-1.0, Fig: https://huggingface.co/stabilityai/stable-diffusion-xl-base-1.0/blob/main/comparison.png

---

> > > > ### Comment · Reviewer_vJhF · 2023-08-21
> > > >
> > > > Thank you for your response.
> > > >
> > > > 1. What are the values of the metrics (e.g., CLIP score) of the proposed model based on SD1.5?
> > > >
> > > > 2. Are $M^{pos(n)}$ and $M^{neg(n)}$ learnable functions of image token $Q$ and text token $K$?

---

> > > > > ### Author Response · Authors · 2023-08-21
> > > > >
> > > > > Thanks for your response.
> > > > >
> > > > > 1. We did not train the model on SD1.5 using our approach. We will verify the model's performance on SD1.5. Futhurmore, we will conduct an evaluation on ControlNet based on SD2.1. However, due to your response being so close to the end date of the rebuttal period (21 Aug 1pm EDT), we cannot guarantee that we will provide the required data before the deadline. In addition, if the evaluation results cannot be released before the end date, please specify any doubts you have about our previous response. In another words, apart from needing more experimental data,  which part would do you feel  that our response did not substantiate our argument *enhancement in controllability via pose/edge/segmentation mask is NOT attributed to any shortcomings in image quality* with the previous rebuttal?
> > > > >
> > > > > 2. No, they are not learnable.

---

> > > > > > ### Comment · Reviewer_vJhF · 2023-08-21
> > > > > >
> > > > > > Thank you for your response.
> > > > > >
> > > > > > 1. My main concern is about the inconsistency between the quantitative metrics in controllability and general metrics like FID/CLIP score/BLIP Score/HPS.
> > > > > >
> > > > > > 2. Could you clarify when $M_{ij}^{pos(n)}$ is set to 1 in equations?

---

> > > > > > > ### Author Response · Authors · 2023-08-21
> > > > > > >
> > > > > > > Thank you for your response.
> > > > > > > 1. Some mentioned metrics  are strongly related to the training data. In terms of image quality, text fit, and controllability, we verified our hypotheses using HPS, ImageReward, and the corresponding discrimination model, respectively. Among them, HPS[1] and ImageReward[2] are metrics specifically designed to verify text-to-image generation models. Overcoming these biases to achieve fairer results is precisely the purpose of proposing new metrics, making them more persuasive than FID, CLIP, or BLIP.  And our model performs the best on these metrics.
> > > > > > >
> > > > > > > 2. We have attention map $A$, and  $A_{ij}  = exp(Q_i, K_j) / softmax(Q,K)$. For each $K_j$, we have a corresponding attention map $A_j$. And the mask is operated in $A_j$: the corresponding area of $Q_i$ in $A_j$ will be 1. Thus:   $M_{ij}^{pos} = 1$, $\forall A_j(Q_i) \cap A_j$
> > > > > > >
> > > > > > >
> > > > > > > [1] Wu X, Sun K, Zhu F, et al. Better aligning text-to-image models with human preference[J]. arXiv preprint arXiv:2303.14420, 2023.
> > > > > > >
> > > > > > > [2] Xu J, Liu X, Wu Y, et al. Imagereward: Learning and evaluating human preferences for text-to-image generation[J]. arXiv preprint arXiv:2304.05977, 2023.

---

### Author Rebuttal · Authors · 2023-08-03

We have organized the comments from the reviewers, and below are several issues that have been highlighted by multiple reviewers.

**Clarification of Benefits and Analysis:**

In the principal section of this submission, metrics such as CLIP, FID, and aesthetic scoring were employed to gauge our model's performance. These metrics often stress the model's capacity to generate images. For example, FID measures how closely the generated images fit the distribution of real images, while the CLIP Score represents the correlation between the generated images and corresponding text, and aesthetic scores gauge the artistic quality. The FID score is highly linked to the training dataset, and due to resource constraints, using only a subset of LAION might decrease diversity, leading to a higher FID.

We emphasize that our work focuses on image controllability while maintaining quality. Thus, additional metrics have been introduced in the appendix for a more comprehensive evaluation of control over image attributes. These supplementary metrics process the images into given control modality signals, then quantify the differences between generated images and provided signals.  For example, we obtain the HED of the generated image, then calculate its L2 loss relative to the conditional HED input. These supplementary metrics process the generated images into the given control modality signals, subsequently quantifying the discrepancies between the generated images and the provided signals. We repeat our evaluation results as Tables 1 and 2 in the attachment. It can be found that our approach is superior to the baseline in controllability across all trained modalities.


**Computational Complexity and Additional Details:**

Alleviating the substantial overhead of multi-modal control is a primary goal. We're happy to clarify specific details about our model's parameters to illuminate this work's contributions.

Our employed SD base model's parameter count is 865.91M, with a single ControlNet having 378.50M parameters. Our fusion network $\mathcal{M}$ consists of only 1.29M parameters per modality, and each ControlNorm adds just 0.31M parameters.
In single modality control using ControlNet, the total parameters are 865.91M + 378.50M. With an increase in modalities, this overhead grows linearly, reaching 865.91M + $n$ × 378.50M where $n$ is the number of modalities.
Our innovative gControlNet handles multiple modalities via a single HyperNetwork efficiently. Without ControlNorm, the parameters count is 865.91M + 378.50M + $n$ × 1.29M, substantially cutting the computational needs in multi-modal contexts and highlighting our approach's efficiency.

$\mathcal{M}$ is a straightforward convolutional network with eight layers and seven SiLU activation functions, with the last layer initialized to zero. Each  $\mathcal{M}$ has 1.29M parameters, and each modality will have a corresponding $\mathcal{M}$, for instance, 3 different modalities in our case result in 3.88M total parameters. It can effectively manage any number of trained modalities but can't handle unknown ones, meaning the model can process subsets and the exact set of training modalities.

Compared with gControlNet without ControlNorm and ControlNet, there will be a $\mathcal{M}$ in the former's training process. This operation might degrade the quality of the generated images without the assistance of ControlNorm. And after passing through $\mathcal{M}$ and gControlNet will there be only a set of control signals injected into the base model while ControlNet or T2I will have multipile set of control signals.  In addition, unlike T2I and ControlNet, where different modalities are trained separately, gControlNet co-trained $\mathcal{M}$ with different modalities.

Codes will be made available for reproduction.

**Ethical Considerations:**

We acknowledge your concern about the potential misuse of our model. The images that our model can generate heavily depend on the training dataset used. Currently, our model's training data strictly complies with open-source licenses and has been carefully selected, providing some degree of safeguard against misuse. In situations where the model generates certain objects (including human being), many details can be easily distinguished. Going forward, we are considering the addition of watermarks to the generated images to prevent unauthorized use, along with the implementation of a monitoring network at the output stage. Similar tools are already widely utilized in projects like Stability / DALL.E2, and we believe our model can also benefit from adopting such measures.

**Conflicting control signals**

When confronted with conflicting signals across different modalities, the model endeavors to interpret each modality coherently. It strives to incorporate all the information conveyed by the different modalities, translating them into conditions for the generated images. The result is a theoretically reasonable representation that balances and harmonizes the conflicting inputs, ensuring that the synthesized images reflect as much of the diverse input data as possible.  However, there is a possibility that one modality may overshadow another if its representation is significantly more dominant or contains richer information. In such cases, the more expressive modality may suppress the other, leading to an unbalanced influence on the generated images. This phenomenon might result in the loss of some nuanced details conveyed by the less dominant modality. Some samples are demonstrated as Fig.1 in the attachment.

---

### Decision · Program_Chairs · 2023-09-21

**Decision:**

Accept (poster)

**Comment:**

This paper introduces a novel idea supported by strong empirical results. The reviewers appreciate the valuable contribution of this study. In preparation for the camera-ready version, we kindly urge the authors to incorporate the feedback provided by the reviewers, which will undoubtedly further refine the quality and impact of the work.